# dia-PASEF data analysis using FragPipe and DIA-NN for deep proteomics of low sample amounts

Vadim Demichev [1,2,3,9 ✉], Lukasz Szyrwiel[1,2,9], Fengchao Yu [4], Guo Ci Teo [4], George Rosenberger [5], Agathe Niewienda[1], Daniela Ludwig[1], Jens Decker[6], Stephanie Kaspar-Schoenefeld[6], Kathryn S. Lilley [3], Michael Mülleder[7], Alexey I. Nesvizhskii [4,8 ✉] & Markus Ralser [1,2]

The dia-PASEF technology uses ion mobility separation to reduce signal interferences and increase sensitivity in proteomic experiments. Here we present a two-dimensional peak-picking algorithm and generation of optimized spectral libraries, as well as take advantage of neural network-based processing of dia-PASEF data. Our computational platform boosts proteomic depth by up to 83% compared to previous work, and is specifically beneficial for fast proteomic experiments and those with low sample amounts. It quantifies over 5300 proteins in single injections recorded at 200 samples per day throughput using Evosep One chromatography system on a timsTOF Pro mass spectrometer and almost 9000 proteins in single injections recorded with a 93-min nanoflow gradient on timsTOF Pro 2, from 200 ng of HeLa peptides. A user-friendly implementation is provided through the incorporation of the algorithms in the DIA-NN software and by the FragPipe workflow for spectral library generation.

[1] Department of Biochemistry, Charité – Universitätsmedizin Berlin, Berlin, Germany. [2] Molecular Biology of Metabolism Laboratory, The Francis Crick Institute, London, UK. [3] Department of Biochemistry and Milner Therapeutics Institute, University of Cambridge, Cambridge, UK. [4] Department of Pathology, University of Michigan, Ann Arbor, MI, USA. [5] Department of Systems Biology, Columbia University, New York, NY, USA. [6] Bruker Daltonics GmbH & Co. KG, Bremen, Germany. [7] Core Facility High-Throughput Mass Spectrometry, Charité – Universitätsmedizin Berlin, Berlin, Germany. [8] Department of Computational Medicine and Bioinformatics, University of Michigan, Ann Arbor, MI, USA. [9]These authors contributed equally: Vadim Demichev, Lukasz Szyrwiel. ✉email: vadim.demichev@charite.de; nesvi@med.umich.edu

High-throughput proteomic experiments are increasingly required for systems biology and biomedical applications. Constrained by the complexity of the proteome, fast proteomics creates substantial analytical challenges. Data-independent acquisition (DIA) techniques have been introduced to improve robustness and reduce missing value rates in large proteomic studies[1,2]. Due to the developments in mass spectrometry (MS) instrumentation and software, DIA methods have recently gained in depth, data consistency, and quantitative accuracy[3]. For example, a recent study demonstrated the quantification of more than 10,000 proteins in single MS runs[4]. Furthermore, significant progress in the analysis of DIA data has allowed it to move to faster gradients and higher flow rates. Higher flow rates provide the benefit of high peak capacities as well as increase column lifetime and chromatographic stability. As a consequence, they simplify very large and longitudinal experiments that were previously difficult to achieve[5–8]. Moreover, while the higher sample dilution did at least initially restrict the proteomic depth[5], these constraints are increasingly mitigated. For example, we have recently demonstrated Scanning SWATH acquisition, which, coupled to 800 μL/min high-flow chromatography, yielded precise quantification of over 2500 proteins in 60 s of active chromatographic gradient, and despite the high flow rates, achieved the quantification of more than 4000 proteins in 5-min high-flow gradients[9]. The gains in throughput to hundreds of samples per day per mass spectrometer give space to new applications, such as exploratory drug screens and clinical studies with high participant numbers. However, the high sample dilution caused by fast chromatographic methods remains a challenge for experiments that require the analysis of low sample amounts, such as single-cell proteomic experiments, in the analysis of micro-biopsies, or deep spatial tissue analysis using laser microdissection[10–12]. These applications call for increased depth, and acceleration also of proteomic experiments that rely on low flow rate chromatography.

An effective strategy to increase the sensitivity of any analytical method is to increase the signal-to-noise ratio. A recent DIA acquisition method, dia-PASEF, utilizes a Trapped Ion Mobility Separation (TIMS)[13] device within the timsTOF Pro mass spectrometer (Bruker Daltonics), to achieve an ion mobility separation of proteomic samples[14]. In dia-PASEF, the ion mobility dimension allows to distinguish signals from peptides that would otherwise be co-fragmented, thus producing cleaner spectra. Important for the analysis of low sample amounts, dia-PASEF can also gain a factor of 2–5 times in sensitivity, depending on the acquisition scheme, by "stacking" precursor ion isolation windows in the ion mobility dimension and thus increasing the effective duty cycle[14].

Here we present a computational strategy for the analysis of ion mobility proteomic data acquired with dia-PASEF. This includes algorithms for two-dimensional (2D) peak picking and a software solution for the processing of the TIMS dimension. To make these accessible to the proteomic community, we incorporated these algorithms in a TIMS module within DIA-NN, an automated, fast, and easy-to-use software suite that employs deep neural networks in DIA data analysis[15]. Moreover, we show that the depth of dia-PASEF experiments is further improved through the generation of optimized spectral libraries from offline-fractionated PASEF (DDA) data with the FragPipe computational platform, using the MSFragger search engine[16,17] coupled to peptide validation, protein inference and false discovery rate (FDR)-based filtering using Philosopher[18]. We show that these developments increase the protein identification performance of dia-PASEF by up to 83%, especially in fast proteomic experiments that use low sample amounts, while simultaneously increasing data consistency as well as quantification accuracy and precision.

## Results

**TIMS module in DIA-NN and generation of spectral libraries.** For the deep neural network-based processing of dia-PASEF experiments, we extract ion mobility-separated data, characterize the quality of peptide-spectrum matches using the agreement between the expected and observed ion mobility values, and assess these quality scores using neural networks (Fig. 1). In contrast to the direct extraction of "profile" ion mobility data as implemented in the Mobi-DIK module for OpenSWATH, the first software that was able to process dia-PASEF data[14], we present an ion mobility module that starts the analysis from 2D-peak-picking, wherein a narrow scanning window is used to find local maxima in the 1/K0 x m/z space, where 1/K0 is the inverse ion mobility (Methods). The subsequent chromatogram extraction is then performed as follows: for each precursor ion and for each of its fragment ions, the most intense peaks are identified within a particular mass threshold (automatically determined or user-defined) and ion mobility threshold ("ion mobility window"; automatically determined based on the alignment between confident identifications and the spectral library).

We speculated that our approach based on 2D-peak-picking should be highly efficient in maximizing the sensitivity and minimizing interferences. The sensitivity promotes high proteomic depth when analyzing low sample amounts and is enabled by fully capturing the signal of each fragment ion in the vicinity of its apex in the process of 2D-peak-picking (Fig. 1). Simultaneously, scanning the 1/K0 x m/z space with a narrow window minimizes the integration of interfering signals originating from other peptides (Fig. 1). This is particularly beneficial for proteomic data recorded with fast chromatographic gradients, wherein separation of peptides in the retention time dimension is limited by the low peak capacity.

Next, once the whole chromatograms are extracted for a specific precursor and its fragments, candidate peaks, featuring signals from at least two different fragments, are identified and scored. The consistency of ion mobility values between fragment ions is taken into account, and "outlier" fragments get lower scores, even if their elution profiles show a high correlation with those of other fragments. This eventually leads to preferential selection of candidate elution peaks with high consistency of ion mobility values across fragments. Furthermore, the deviation of observed fragment ion mobilities from the reference precursor ion mobility value obtained from the spectral library is likewise taken into account. Ultimately, these scores are fed into the ensemble of deep neural networks to assign confidence scores to precursor-spectrum matches. This way the performance of the ion mobility module benefits from the high flexibility and robustness of the neural network classifier as introduced with DIA-NN[15], which further facilitates the identification of peptides with high confidence from complex data[15]. Finally, signals with deviating ion mobility values are excluded during the quantification of peptides, thus improving quantitative accuracy (Fig. 1). Eventually, we assembled the algorithms in an "ion mobility module", and, to provide an easy-to-use software solution, integrated them into the DIA-NN software suite[15].

We speculated that further gains in dia-PASEF performance can be obtained with an adapted workflow for the generation of specific spectral libraries. We extended our earlier work[17] and optimized a data analysis workflow (see Methods) in FragPipe, for generating spectral libraries from fractionated PASEF data in the format directly compatible with the ion mobility-enabled version of DIA-NN.

**Identification performance.** To benchmark the performance of the TIMS module in DIA-NN in combination with FragPipe-

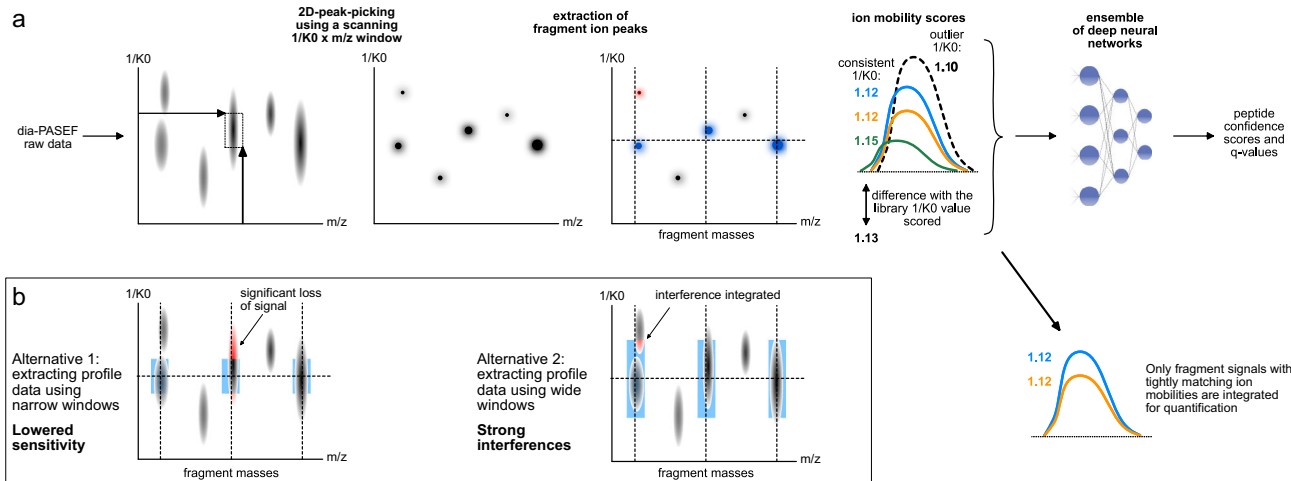

**Fig. 1 A concept for processing of proteomic trapped ion mobility data. a** Our dia-PASEF data processing workflow starts with 2D-peak-picking using a narrow scanning window. Chromatogram extraction is then performed, wherein for each precursor or fragment ion, only peaks within certain m/z and ion mobility thresholds from the expected values are used. Expected values are indicated here with dotted lines, peaks discarded due to m/z thresholding are indicated in gray, and a peak discarded due to only ion mobility thresholding is in red. Observed inverse ion mobility values (1/K0) are compared between different fragment ions (extracted chromatographic elution profiles and apex 1/K0 values of which are indicated with different colors) as well as to the reference library 1/K0 value (here: 1.13), to score putative peptide-spectrum matches. Fragments with outlier ion mobility values (here: black—signal from another peptide, green—signal mildly affected by interference) are assigned lower scores. The resulting data are analyzed by an ensemble of deep neural networks, used to distinguish true and false signals. Signals with deviating ion mobility values are also filtered out to increase quantification accuracy. **b** In contrast to the 2D-peak-picking introduced herein, direct extraction of chromatograms from the profile data could potentially be used. In this case, if extracting profile data with narrow windows (here: in blue), for example, the same size as used by the 2D-peak-picking algorithm, a significant proportion of ion signal can be lost (example highlighted in red) due to an imperfect match between theoretical and empirical m/z or 1/K0 values. If extracting with wide windows, more interfering signals would be integrated (example highlighted in red), increasing the complexity of the data and hampering correct identification and accurate quantification of peptides.

generated libraries, we first reprocessed the reference dia-PASEF data[14], wherein a HeLa tryptic digest was acquired with different injection amounts (10–100 ng) and acquisition schemes using a nanoLC (EASY-nLC 1200, Thermo Fisher) separating peptides with an active LC gradient of 95-min. Data were recorded using a 25% duty cycle scheme or a "standard" scheme, or using pre-formed microflow gradients (Evosep One microflow system, Evosep) with different chromatographic gradient lengths (200 ng, 5.6–21 min; Fig. 2a).

In the experiments using nanoflow chromatography, the FragPipe-generated spectral library built from 24 fractionated DDA PASEF runs and filtered at 1% protein and peptide FDR contained 9991 proteins and 161,325 peptides (see Table 1 and Methods). The analysis with the herein developed ion mobility module yielded 28–56% improvement in protein numbers. This performance advantage led to over five-fold improvement in sensitivity, in the sense that our workflow quantified more proteins from 10 ng of HeLa peptides than the previous workflow did from 50 ng.

Many of the algorithms consolidated in DIA-NN had originally been conceived to maximize the performance of fast proteomic experiments[15]. Indeed, the software suite achieves high confidence in peptide identification with fast acquisition schemes such as Scanning SWATH, even with high flow rate chromatographic gradients as fast as 30 s to 5 min[9,15]. High flow rate chromatography is however not the method of choice for low sample amounts, as the higher sample dilution due to high volumes needs to be partially compensated by higher injection amounts[6]. Thus, we sought to evaluate the performance of our workflow on fast dia-PASEF data[14], and reanalyzed raw data recorded from HeLa cells with microflow chromatography using the Evosep One system[19]. The spectral library built with FragPipe from 24 DDA PASEF runs contained 8201 proteins,

98,485 peptides, and 145,875 precursor ions (Table 1). Our analysis quantified on average 5323 unique proteins from 200 ng of HeLa digest analyzed with a 5.6-min gradient (200 SPD Evosep One method, SPD = samples per day), an 83% gain compared to the original values[19]. Illustrating that higher proteomic depth can be exploited to accelerate proteomics experiments, our results show that the 5.6-min gradients yielded a number of quantified proteins 10% greater than that reported for the longer 21-min gradient (60 SPD method) in the original publication[14]. In all benchmarks, the gains achieved by our pipeline originated from the detection of higher numbers of medium- and low-abundant proteins (Supplementary Fig. 4).

To illustrate how the different algorithms contribute to the gains in proteomic depth, we sequentially disabled the individual algorithms of the ion mobility module as well as evaluated the impact of a FragPipe library (Supplementary Figs. 1 and 2). We observed that each of the core TIMS algorithms (2D-peak-picking, chromatogram extraction using an ion mobility window, and scoring of peptide-spectrum matches based on the ion mobility information) contributed to the performance gains, with all of them together yielding between 849 and 1177 extra proteins quantified, depending on the gradient length in the Evosep One experiments, respectively. The FragPipe-generated spectral library contributed between 598 and 740 extra protein identifications. Furthermore, we implemented a module in DIA-NN for direct chromatogram extraction from profile dia-PASEF data, as described previously[14], using the OpenSWATH code base as a reference for our implementation. We observed that 2D-peak-picking outperforms the profile extraction, with the performance difference being higher for shorter gradients, which are characterized by higher interference levels. We further benchmarked the effect of the neural network module in DIA-NN, and

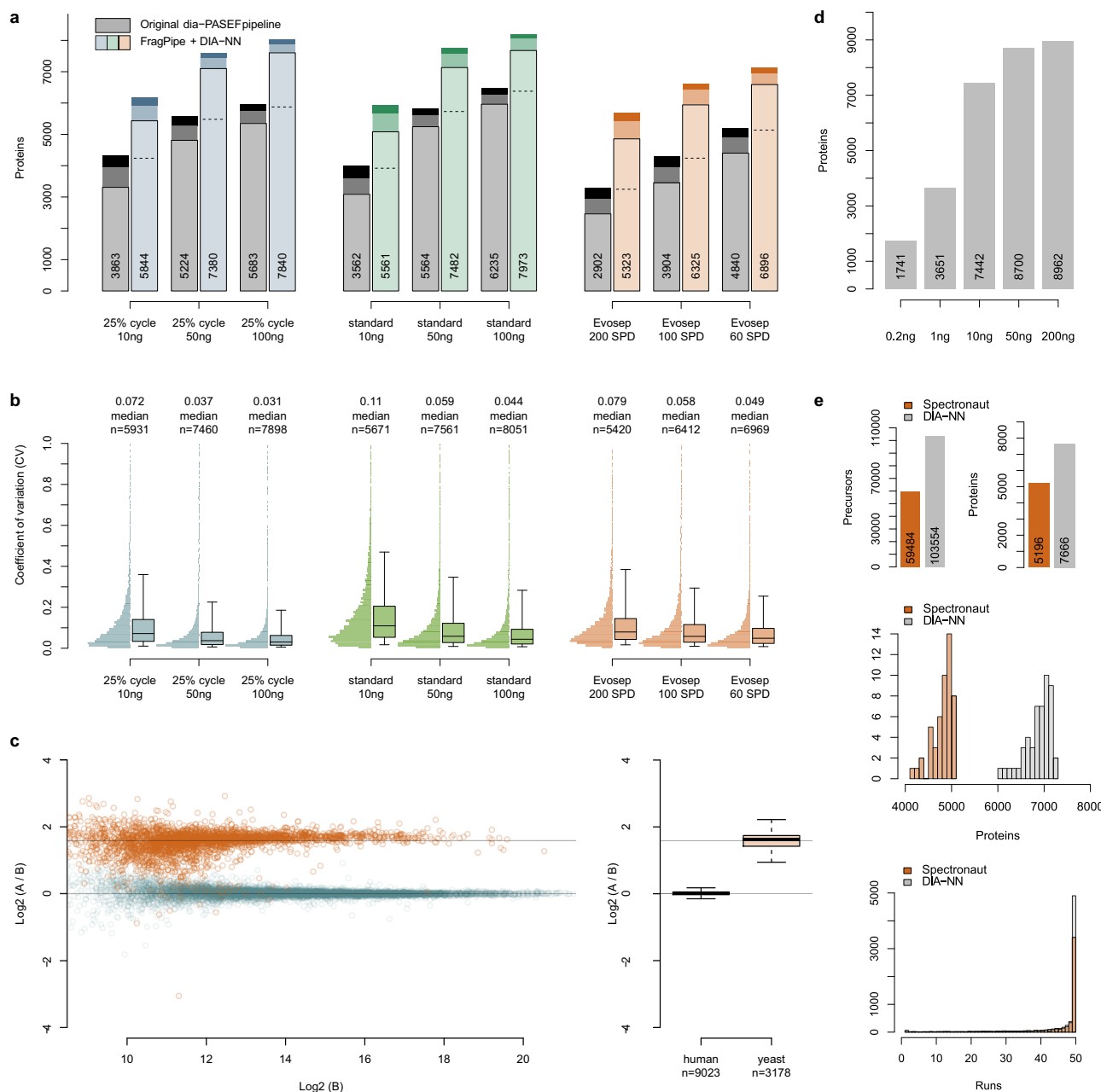

**Fig. 2 Protein detection and quantification performance. a** Number of quantified proteins for different injection amounts and instrument settings. Numbers of proteins detected in 1, 2, or all 3 injection replicates for each dataset (nanoflow 25% duty cycle scheme and standard scheme; Evosep 200, 100, and 60 samples per day (SPD) methods) are shown with different color shades, average numbers are indicated. Numbers reported by the original dia-PASEF workflow are shown in gray[14]. The numbers of proteins detected by both workflows are indicated with dashed horizontal lines. **b** Coefficients of variation (CV) distributions for the same datasets. The boxes correspond to the interquartile range, with the median indicated, and the whiskers extend to the 5–95% percentiles. **c** Quantification accuracy of dia-PASEF data analyzed with the new software workflow. We reanalyzed previously recorded data[14], generated by spiking a yeast digest into a HeLa digest (200 ng) in different proportions (A, 45 ng, and B, 15 ng) and analyzed in triplicates using a 90-min nanoLC gradient. The runs were processed using a spectral library created with FragPipe. Horizontal lines indicate the expected ratios. On the boxplot, the boxes correspond to the interquartile range, with the median indicated, and the whiskers extend by a 1.5× interquartile range. Expected ratios are indicated with gray lines. **d** Analysis of a dilution series acquired on timsTOF Pro 2, a second-generation dia-PASEF-capable mass spectrometer, using a 93-min 300 nL/min gradient and a pre-column (Methods). Average protein numbers for triplicate injections after filtering at 1% run-specific protein q-value are shown. **e** Comparison of the performance of DIA-NN (gray) and Spectronaut (orange) on the leukemia dataset[21]. Total numbers of precursors and proteins (top), protein ID numbers distributions, and consistency of protein detection (bottom) are compared. The y-axis on the histograms represents the counts.

it likewise showed the greatest advantage for shorter gradients, in line with our previous[15] observations (Supplementary Fig. 2).

We validated the identification performance using a two-species benchmark strategy[4], in dia-PASEF data with varying levels of complexity (Evosep One dataset), confirming that the

FDR and hence the identification numbers reported by the software are conservative (Supplementary Fig. 3). Using the same benchmark, we also compared our workflow to Spectronaut[20], another dia-PASEF-capable software. We report a roughly two-fold gain in terms of precursor numbers detected using longer

**Table 1 Spectral libraries.**

| Library | Ions | Peptides | Proteins | Genes |
|---|---|---|---|---|
| HeLa, nanoflow | 260,785 | 161,325 | 9991 | 9973 |
| HeLa, Evosep One | 145,875 | 98,485 | 8201 | 8187 |
| HeLa, two-organism | 361,555 | 224,597 | 10,353 | 10,332 |
| HeLa, two-organism, filtered | 360,458 | 223,907 | 10,350 | 10,331 |
| Yeast, two-organism | 134,148 | 79,860 | 5134 | 5132 |
| Yeast, two-organism, filtered | 133,351 | 79,337 | 5113 | 5113 |

gradients. The observed advantage is consistent with the advantage demonstrated over OpenSWATH: for example, using the library by Meier et al.[14] and analyzing the Evosep 60 SPD runs, our software workflow identifies on average 46,497 precursors, compared to 26,348 reported by OpenSWATH from the same data. We noticed that at a shorter 5.6-min gradient both the performance of Spectronaut and the accuracy of FDR values reported by it further dropped significantly, similarly to what we observed with 20–30 min non-TIMS methods previously[15]. Our software, in contrast, showed robust performance and reliable FDR for all the runs considered.

We further compared the software to Spectronaut using a recent dia-PASEF analysis of 50 chronic lymphocytic leukemia (CLL) samples, recorded with a 100-min 400 nL/min gradient on timsTOF Pro, acquired as part of a larger multi-omics study, aimed at identifying the determinants of the proteome variation in CLL[21]. Here we report a 74% gain on the precursor level and 48% gain on the protein level (Fig. 2c). In this benchmark, Spectronaut was run in directDIA (library-free) mode by the authors of the study, due to the lack of an offline-fractionated pooled sample analysis via DDA, and hence lack of a project-specific spectral library. Likewise, we also performed the analysis using the library-free capabilities of DIA-NN described previously[15], wherein a spectral library is created from DIA data directly, and the data is then analyzed with this library, as was first proposed for the DIA-Umpire workflow[22].

In addition, we benchmarked our software workflow for use with the recently introduced timsTOF Pro 2, the second-generation TIMS-capable[13] instrument. Here the analysis was also performed in library-free mode, due to the lack of a project-specific library. To the best of our knowledge, this is the first performance benchmark of dia-PASEF on this mass spectrometer. We acquired a dilution series of tryptic peptides generated from HeLa cells, measuring 0.2–200 ng on an Ultimate 3000 nanoLC equipped with a trap column, and separated the peptides using a 93-min 300 nL/min chromatographic gradient (Fig. 2e). We report the quantification of 8962 proteins in a single injection from a 200 ng HeLa digest, as well as 7442 proteins from a 10 ng HeLa digest. Compared to the original performance of dia-PASEF on the first generation timsTOF Pro[14], this is a gain of +93% in protein IDs, and +19% in comparison to the numbers reported previously for a 10× higher injection amount[14]. Of note, we measured 3651 proteins from 1 ng of HeLa Extract, almost matching the performance seen previously[14] for 10 ng. We also note that the numbers obtained here are still limited by the use of a regular nanoLC with a relatively high flow rate and the use of a pre-column, which indicates that further increases in depth are possible.

**Quantification performance**. We next evaluated the quantification performance of our data analysis strategy. Although the increase in proteomic depth means more low-abundant peptides are quantified, the overall parameters of robustness and quantification precision were improved. Most notably, we obtained high data completeness, which renders the workflow attractive for the

application of machine learning methods in the analysis of large sample series, which perform best on consistent data. Data completeness ranged from 94% for the 200 SPD Evosep One benchmark to 98% for 100 ng 25% cycle injections analyzed with a 95-min gradient. Moreover, expressed as coefficient of variation (CV), the quantification values were shown to be precise. The median CVs of all datasets did not exceed 11% (Fig. 2b). For datasets with higher injection amounts, we obtained encouragingly low CV values with a median CV of 3.1% for the nanoflow, 25% cycle, 100 ng injection experiment. Notably, a median CV value below 8% was achieved for the fast 200 samples per day Evosep One method. Keeping in mind that the technical variation in the Evosep One benchmarks here reflects not just the variability introduced by chromatography and mass spectrometer performance, but also that due to peptide purification via solid-phase extraction in the filter tips[19], this result demonstrates that our workflow facilitates precision proteomics even with low sample amounts analyzed at high throughput.

Finally, we assessed the quantification accuracy of our workflow. Accuracy is an equally important concept as CV values, and reflects how well quantitative ratios are preserved by the LC-MS and the subsequent data processing workflow. Here we used the two-organism benchmark data (Fig. 2c), wherein a yeast (*Saccharomyces cerevisiae*) tryptic digest was spiked in different concentrations (45 ng, sample A, and 15 ng, sample B) into a human cell line (HeLa) tryptic digest (200 ng) and analyzed with a 90-min nanoLC gradient on a Bruker nanoElute LC system[14]. The human and yeast spectral libraries built with FragPipe from 24 (HeLa) and 48 (Sc) PASEF runs contained 10,353 and 5134 proteins, respectively (Table 1). On average, we report the quantification of 12,225 unique proteins per run for sample A and 11,859 for sample B. The identification of yeast proteins in this benchmark is particularly challenging for the low-concentration sample B (only 15 ng of yeast digest), and thus the numbers of valid A:B ratios reflect the ultimate sensitivity of the workflow. We report 3178 valid A:B ratios for yeast proteins, out of which 2937 proteins were quantified in at least two replicates for each sample, A and B, more than doubling the previously reported number of 1394 for the same experiment[14]. Our data analysis workflow also increased the quantification accuracy, with visibly less grossly incorrect A:B ratios for yeast proteins. The numbers of human protein ratios were moderately higher (9023, with 8924 quantified in at least two replicates for each sample, in comparison to 7697 originally reported). For these, a quantification precision of 4.2% median CV was observed. In total, 12,388 proteins were quantified in the whole experiment.

## Discussion

In summary, we present computational concepts for 2D-peak-picking and software developments for the analysis of proteomic ion mobility data with deep neural networks, as well as the generation of optimized spectral libraries, with both these workflow elements contributing to a gain in performance. We report substantial gains of up to 83% in proteomic depth as well as improved quantification accuracy in the analysis of trapped ion mobility data in fast and conventional proteomic experiments, especially in experiments in which low sample amounts are separated with fast microflow chromatography. To make these developments accessible to the community, we have implemented an ion mobility module in the DIA-NN software suite[15], and have augmented the FragPipe workflow[16–18] for the automated generation of specific spectral libraries. We further provide an integrated version of DIA-NN within FragPipe, allowing for convenient execution of the whole pipeline within a single user interface.

## Methods

**Spectral library generation in FragPipe**. We used FragPipe computational platform (version 15) with MSFragger[16,17] (version 3.2), Philosopher[18] (version 3.4.13), and EasyPQP (version 0.1.9) components to build spectral libraries. Peptide identification from tandem mass spectra (MS/MS) was done using the MSFragger search engine, using either raw (.d) files (Evosep and nanoflow HeLa dataset) or MGF files (two-organism dataset) as input. Protein sequence databases *H. sapiens* (UP000005640) or *S. cerevisiae* (UP000002311) from UniProt (reviewed sequences only; downloaded on February 15, 2021) and common contaminant proteins, containing in total 20421 (*H. sapiens*) and 6165 (*S. cerevisiae*) sequences were used. Reversed protein sequences were appended to the original databases as decoys. For the MSFragger analysis, both precursor and (initial) fragment mass tolerances were set to 20 ppm. Spectrum deisotoping[23], mass calibration, and parameter optimization[24] were enabled. Enzyme specificity was set to "strict-trypsin" (i.e., allowing cleavage before Proline), and either fully enzymatic peptides were allowed. Up to two missed trypsin cleavages were allowed. Isotope error was set to 0/1/2. Peptide length was set from 7 to 50, and peptide mass was set from 500 to 5000 Da. Oxidation of methionine, acetylation of protein N-termini, −18.0106 Da on N-terminal Glutamic acid, and −17.0265 Da on N-terminal Glutamine and Cysteine were set as variable modifications. Carbamidomethylation of Cysteine was set as a fixed modification. Maximum number of variable modifications per peptide was set to 3.

For each analysis, the MS/MS search results were further processed using the Philosopher toolkit[18]. First, MSFragger output files (in pepXML format) were processed using PeptideProphet[25] (with the high–mass accuracy binning and semi-parametric mixture modeling options) to compute the posterior probability of correct identification for each peptide to spectrum match (PSM). The resulting output files from PeptideProphet (also in pepXML format) were processed using ProteinProphet[26] to assemble peptides into proteins (protein inference) and to create a combined file (in protXML format) of high confidence proteins groups and the corresponding peptides assigned to each group. The combined protXML file was further processed using Philosopher Filter module as follows. Each identified peptide was assigned either as a unique peptide to a particular protein (or protein group containing indistinguishable proteins) or assigned as a razor peptide to a single protein (protein group) that had the most peptide evidence. The protein groups assembled by ProteinProphet, with the probability of the best peptide used as a protein-level score[27], were filtered with 1% protein-level FDR using the picked FDR strategy[28], allowing unique and razor peptides. The final reports were then generated and filtered at each level (PSM, ion, peptide, and protein) using the 2D FDR approach[29] (1% protein FDR plus 1% PSM/ion/peptide-level FDR for each corresponding PSM.tsv, ion.tsv, and peptide.tsv files).

Finally, PSM.tsv files, filtered as described above, along with the spectral files (original MGF files, or uncalibrated MGF files created by MSFragger when raw.d files were used as input to MSFragger) were used as input to EasyPQP for the generation of the consensus spectrum libraries. In doing so, a peptide's retention times in each fraction were non-linearly aligned (lowess method) by EasyPQP to a common iRT scale using the extended HeLa iRT calibration peptide set[30]. Peptide's ion mobility values in each run in the dataset were aligned to that from one of the runs in the dataset automatically selected as a reference run. The library was additionally filtered to keep only peptides contained in the Philosopher-generated peptide.tsv report file, ensuring that the final spectral library was filtered to 1% protein and 1% peptide-level FDR.

**2D-peak-picking algorithm in DIA-NN and subsequent chromatogram extraction**. The 2D-peak-picking algorithm in DIA-NN consists of two steps: identification of local signal maxima in the 2D (1/K0 x m/z) spectrum, and selecting, out of multiple adjacent local maxima (often occurring due to noise in the data), those which are likely to be the best representative of the actual peptide signal. The initial identification is done by selecting local maxima using summing signals within a scanning window. Specifically, the window is represented by the following tolerances: (i) maximum ion mobility tolerance is expressed as a number of TOF scans, as recorded in the dia-PASEF acquisition, and is set to 10[frame scan range/900]/[frame 1/K0 range]); (ii) maximum m/z tolerance is expressed as a number of mass bins, as recorded in the dia-PASEF acquisition, and is set to 2; (iii) within this window, only peaks for which the quantity [ion mobility deviation from window center] + 2[mass deviation][frame scan range/900]/[frame 1/K0 range] is less than the ion mobility tolerance (set by (i)) are being summed. Each local maximum (i.e., the scanning window position which results in the highest sum of signals, across all neighboring positions) is assigned an "intensity" score, which is the respective sum of signals within the scanning window. During the second filtering step, candidate local maxima are discarded if there are any more intense maxima within their scanning window with a mass difference of no more than one mass bin. The remaining local maxima are reported as the candidate peaks. For these, three values are stored: the m/z and the 1/K0, which correspond to the scanning window position, as well as the sum of the signals within the window, representing the peak intensity. A special procedure is used if several identical dia-PASEF frames are acquired within the dia-PASEF cycle, which is the way the highest sensitivity was achieved for the low injection amounts[14]. In this case, the respective m/z windows are matched together, and the profile data is summed for these windows, before carrying out the 2D-peak-picking.

We note that the size of the 2D scanning window in the ion mobility dimension is kept the same regardless of the ramp time and the respective number of scans in the frame, and has been chosen based on the observed peak widths in the data we have examined so far. However, too low or too high ramp times might significantly affect resolution in the ion mobility dimension, so future acquisition schemes might benefit from further optimization of the 2D scanning window size.

The chromatograms are extracted by querying a particular m/z value of interest, corresponding to a precursor or fragment ion, against each spectrum. The peaks in each spectrum are stored ordered by mass. Binary search is used to find a peak that falls within the specified mass tolerance from the query m/z. DIA-NN then finds the first and last peaks in the ordered spectrum, which lie within the mass tolerance bounds. Out of the peaks in-between, peaks within the IM window (with the IM tolerance set by DIA-NN automatically, based on the observed IM deviations of peptides identified during the calibration stage of the search) from the predicted IM value (obtained using aligning observed vs the library IM values for the peptides identified during the calibration stage) are selected. Out of these, the peak with the highest intensity is reported.

**Spectral library processing in DIA-NN**. For the two-proteome human-yeast benchmark, the HeLa library was filtered to only include peptides present in the in silico tryptic digest of the human database and exclude peptides present in the tryptic digest of the yeast database and vice versa, by generating the annotation of the library using the "Reannotate" function in DIA-NN and discarding peptides matched to one or more proteins of the other species.

**DIA software configuration and dia-PASEF data processing**. The TIMS module was incorporated in DIA-NN (version 1.8.1), which was used for the benchmarks and was operated with maximum mass accuracy tolerances set to 10 ppm for both MS1 and MS2 spectra. Protein inference was disabled for analyses using DDA-based spectral libraries, to use the protein groups therein. The --relaxed-prot-inf option was used for library-free processing of the leukemia dataset and the HeLa dilution series on timsTOF Pro 2, as this option implements a protein grouping strategy similar to the one used in FragPipe. Library generation was set to "IDs, RT and IM profiling". MBR was enabled for the two-species human-yeast benchmark. When reporting protein numbers and quantities, the Protein.Group column in DIA-NN's report was used to identify the protein group and the PG.MaxLFQ column (calculated using the MaxLFQ algorithm[31]) was used to obtain the normalized quantity. PSM tables (PSM.tsv files generated by Philosopher) contain accession numbers of all mapped proteins for each identified peptide, and this information was used to identify proteotypic peptides. In the benchmark for the numbers of unique proteins from the original dia-PASEF publication (Supplementary Fig. 2), the "Genes" column was used to count unique proteins (as gene products identified and quantified using proteotypic peptides only). For this, proteotypic peptides were annotated as such using the "Reannotate" option. Quantification mode was set to "Robust LC (high precision)". All other settings were left default. Following previously published recommendations[32], and similarly to the previous dia-PASEF workflow[14], the software output was filtered at precursor q-value <1%. Global protein q-value <1% filter was also applied to all benchmarks, except for the HeLa dilution series on timsTOF Pro 2, wherein data were filtered for run-specific protein q-value <1%.

Of note, one of the two-proteome human-yeast benchmark files (200113_AUR_dia-PASEF_HY_200ng_15ng_90min_Slot1-5_1_1636.d) could not be read correctly due to data corruption, with all frames (i.e., dia-PASEF scans) from 55,877 onwards being discarded, which might have resulted in the benchmark results being very slightly worse.

For the FDR accuracy benchmark using a human—*A. thaliana* spectral library, Spectronaut 14.3.200701.47784 was run using default settings, except protein q-value filtering was set to 1 (i.e., 100%), and PTM localization was disabled. The Spectronaut 14.4 analysis of the leukemia dataset in directDIA mode was downloaded from the PXD022216 repository.

**FDR validation benchmark using a two-species human-Arabidopsis library**. The library[4] (repository PXD013658, file "HumanThalianaDDAOnly (two-species FDR test).xls") was filtered to only include unmodified peptides or peptides with carbamidomethylated cysteines. DIA-NN was set to replace all spectra and retention times in this library with in silico predicted ones, as well as generate in silico reference ion mobility (1/K0) values. This was done to eliminate any potential bias in spectral quality between human and plant peptides in this experimental library. "Precursor FDR" filter was set to 10%.

The experimental FDR was then determined for each of the Evosep One runs with different gradient lengths, by counting *A. thaliana* precursor/protein calls. Specifically, the FDR was determined as

$$\text{Experimental FDR} = [A_{id}/(A_{id} + H_{id})][(A_{lib} + H_{lib})/A_{lib}]\pi 0, \quad (1)$$

where $A_{id}$ is the number of calls of *A. thaliana* precursors or proteins at a particular score threshold, $H_{id}$ the calls of human precursors or proteins, and $A_{lib}$, $H_{lib}$ the respective numbers of precursors or proteins in the library. The $\pi_0$ ("prior probability of incorrect identification", also known as PIT - Percentage of Incorrect

Targets) correction factor[33,34] was calculated as

$$\pi 0 = (A_{lib} + H_{lib} - 0.95\, H_{id}^{(0.05)})/(A_{lib} + H_{lib}), \qquad (2)$$

where $H_{id}^{(0.05)}$ is the number of human precursor or protein calls at $q$-value = 0.05. The numbers of precursors/proteins were obtained based on filtering the library for precursors within the mass range 400.0–1000.0 m/z (the range sampled in the Evosep One dia-PASEF runs under consideration). The experimentally detected FDR was then plotted (Supplementary Fig. 3) against the precursor/protein FDR estimates reported by the software workflow.

The respective analysis for Spectronaut was performed using in silico prediction with DIA-NN 1.8 and the same formulas for experimental FDR evaluation.

**Comparison of protein numbers to the OpenSWATH workflow**. Average protein identification numbers obtained previously[14] for the same experiments using the OpenSWATH workflow were calculated based on the respective pyprophet reports, which were downloaded from the PRIDE repository with identifier PXD017703. To enable direct comparisons with the OpenSWATH workflow, only proteins uniquely identified using proteotypic peptides were counted for the respective benchmarks, to match the filtering strategy applied for OpenSWATH previously[14].

**HeLa culture, sample preparation, and acquisition**. HeLa cells (ATCC) were cultured in Basal Iscove media (Biochrom) supplemented with 10% fetal calf serum (Biochrom) and 1% penicillin-streptomycin (Biochrom) at 37 °C with a humidified atmosphere of 5% $CO_2$. After three passages, cells were treated with Trypsin/EDTA (Biochrom) and centrifuged at 200× g for 5 min. The pellet was once washed with Dulbecco's Phosphate Buffered Saline.

We have tested three protocols varying in the protein extraction, digestion, and peptide purification methods (P1–P3).

P1 (SDC protocol): $2 \times 10^5$ cells were resolved in 75 μL Lysis-Reduction-Alkylation Buffer, 10 mM Tris-2(2-carboxyethyl)-phosphine-hydrochloride-solution (TCEP, Merck), 40 mM 2-Chloroacetamide (CAA, Merck), 100 mM Tris pH 8.5 and 1% Sodium Deoxycholate (SDC, Merk).

P2 (Urea protocol): $2 \times 10^5$ cells were resolved in 75 μL Lysis-Reduction-Alkylation Buffer, 40 mM CAA (Merck), 100 mM Tris pH 8.5 and 8 M Urea, 10 mM TCEP (Merck).

For P1 and P2, cells were boiled at 95 °C for 5 min in a thermomixer at 800 rpm. Afterward, the cells were sonicated for 20 min on ice in an UltraSonic Bath (Branson). The lysate was diluted 1:10 with HPLC-grade water for P1, and 1:5 with a dilution buffer (10% Acetonitrile (ACN) v/v, 25 mM Tris pH 8.5) for P2.

For P1 and P2, the lysates were digested at 37 °C using Trypsin/LysC (Promega) in a 1:50 (enzyme:protein) ratio. After overnight the digest was acidified to a final concentration 0.5% with trifluoroacetic acid (TFA, Thermo).

Peptides were purified using StageTips (C18 disk, Affinisep), activated with 50 μL of methanol washed/centrifuged in a two-step procedure, one with 50 μL 80% ACN/0.1% FA, and the second with 50 μL 0.1% FA. After the sample load, the tips were washed with 0.1% FA and peptides were eluted with 30 μL 80% ACN/0.1% FA (each time centrifuged for 5 min at 500 × g).

P3 (bulk): $1 \times 10^7$ cells were resolved in a 1 mL lysis buffer (8 M urea, 100 mM ammonium bicarbonate (ABC)), incubated for 30 min at room temperature and 800 rpm in thermomixer (Eppendorf) and the sample was centrifuged for 20 min at 20,817 × g. The lysate was reduced with dithiothreitol (DTT, final concentration 1 mM) for 30 min at room temperature and alkylated with iodoacetamide (IAA, final concentration 5 mM in dark). The sample was diluted 1:3 with 100 mM ABC and digested with trypsin: protein (1:50, Promega) at 37 °C overnight. Peptides were acidified with TFA (final concentration 1%) and purified with Sep-Pak C18 Cartridge, 50 mg Sorbent (Waters). Eluates were dried in a vacuum concentrator (Eppendorf). Samples were resolved in 30 μL 2% ACN/0.1%TFA.

Digests obtained with P1–P3 as well as a commercial HeLa tryptic digest (Thermo, 88329) were analyzed. A dilution series was then obtained for P1, with injection amounts ranging from 0.2 to 200 ng.

The tryptic digests were injected using the autosampler on a pre-column (PepMap C18, 5 mm × 300 μm × 5 μm, Thermo Scientific) with 2% ACN/water (v/v) containing 0.1% TFA at a flow rate of 20 μL/min for 5 min and separated on the 25 cm HPLC column equipped with emitter (Aurora series, CSI, 25 cm × 75 μm ID, 1.6 μm C18, IonOpticks) operating at 50 °C controlled by the Column Oven PRSO-V1-BR (Sonation), using UltiMate 3000 (Thermo Scientific Dionex). The UPLC systems were coupled with TIMS quadrupole time-of-flight instrument (timsTOF Pro 2, Bruker Daltonics) and samples were measured in dia-PASEF mode. The column emitter was installed in the nano-electrospray source (CaptiveSpray source, Bruker Daltonics) and the source parameters were: 1500 V of Capillary voltage, 3.0 L/min of dry gas, and 180 °C of dry temperature. The analytical column flow was set to 300 nL/min and the mobile phases water/0.1% FA and ACN/0.1% FA (A and B, respectively) were applied in the linear gradients starting from 2% B and increasing to 17% in 87 min, followed by an increase to 25% B in 93 min, 37% B in 98 min, 80% B in 99–104 min, the column was equilibrated in 2% B by next 15 min (all % values are v/v, Water and ACN solvets were purchased from Thermo Scientific Price LC-MS grade). For calibration of ion mobility dimension, three ions of Agilent ESI-Low Tuning Mix ions were selected (m/z [Th], 1/K0 [Th]: 622.0289, 0.9848; 922.0097, 1.1895; 1221.9906, 1.3820). For sample injection of more than

50 ng (50 ng or less), the dia-PASEF windows scheme was ranging in dimension m/z from 400 (396) to 1200 (1103) and in dimension 1/K0 0.7 (0.6)–1.43 (1.3), with 32 × 25 Th (59 × 12 Th) windows with Ramp Time 100 ms (166 ms). The window schemes are illustrated in Supplementary Fig. 5. For injections of 50 ng and below the mass spectrometer was further operated in low sample injection mode.

**Reporting summary**. Further information on research design is available in the Nature Research Reporting Summary linked to this article.

## Data availability

The mass spectrometry proteomics data (HeLa dilution series) have been deposited to the ProteomeXchange Consortium (http://proteomecentral.proteomexchange.org) via the PRIDE partner repository[35] with the dataset identifier PXD029836. Previously generated data used in this study are further available from ProteomeXchange Consortium repositories with identifiers PXD017703, PXD022216, and PXD013658. The UP000005640 (human) and UP000002311 (yeast) sequence databases used in this work are available from the UniProt repository. Software output reports, spectral libraries, PSM tables, logs, and the pipeline configuration file were deposited to an OSF (Open Science Framework) repository https://doi.org/10.17605/OSF.IO/8EPQH.

## Code availability

The TIMS module is integrated into DIA-NN 1.8.1, available for download from https://github.com/vdemichev/DiaNN. FragPipe is available for download from https://fragpipe.nesvilab.org/. MSFragger can be downloaded from http://msfragger.nesvilab.org/ or directly from FragPipe. The input data and scripts used to generate the figures and numbers reported in this manuscript were deposited to an OSF (Open Science Framework) repository https://doi.org/10.17605/OSF.IO/8EPQH.

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

## Acknowledgements

We thank Sven Brehmer and Nagarjuna Nagaraj (Bruker) for helpful discussions and their advice on the analysis of dia-PASEF data. We thank Dmitry Avtonomov for his help with FragPipe development. This work was funded in part by NIH grants R01-GM-094231, U24-CA210967 and U24-CA271037 (to A.I.N.), the BBSRC (BB/N015215/1 to M.R., BB/N015282/1 to K.S.L.), the Francis Crick Institute, which receives its core funding from Cancer Research UK (FC001134), the UK Medical Research Council (FC001134), and the Wellcome Trust (FC001134 and IA 200829/Z/16/Z to M.R.), the ERC under grant agreement ERC-SyG-2020-951475 (to M.R.), and the Ministry of Education and Research (BMBF), as part of the National Research Node "Mass spectrometry in Systems Medicine" (MSCoreSys), under grant agreements 031L0220A (to M.R.) and 161L0221 (to V.D.).

## Author contributions

Algorithms and data analysis: V.D., F.Y., G.C.T., A.I.N., G.R., J.D., S.K.-S.; experimental design: V.D. and L.S.; sample preparation: A.N., D.L.; LC-MS: L.S.; raising funding: M.R., A.I.N., V.D., K.S.L.; supervision: M.R., A.I.N., V.D., M.M., K.S.L.; initial draft: V.D., M.R., A.I.N., L.S.; writing: all authors approved the final manuscript.

## Funding

## Competing interests

J.D. and S.K.-S. are employees of Bruker Daltonics. The remaining authors declare no competing interests.
