## [Peer Review File · Nature Communications]

Reviewers' Comments:

Reviewer #1:

Remarks to the Author:

In this manuscript, Demichev et al. present software algorithms and optimized spectral libraries that facilitate processing of ion mobility information in diaPASEF data. The authors extended their previously developed DIA-NN framework by adding a 2D peak picking algorithm and extracting several IM related information as neural network inputs. The authors also prepared specific spectral libraries suitable for use with diaPASEF data analysis. The authors benchmarked their IM-DIA-NN algorithm on previously published datasets, and showed generally improved performance over the original diaPASEF report, with a significant increase in protein identification in certain cases (up to 72%). The authors also studied the quantitation accuracy, and looked into effects of semi-tryptic peptides.

Increasing proteomics throughput and using lower sample amounts without compromising proteomic depth is clearly one of the key directions for future proteomics, and this manuscript reports significant gains in protein identification in certain small samples (similar number of proteins ids from a 5-min gradient as the original 12-min gradient). Nonetheless, I am unable to recommend this paper for publication in Nature Communications based on the following grounds.

First, the DIA-NN neural network framework has been developed previous by the authors, and what's new here is the addition of a 2D peak picking algorithm (and matching IM values). While this algorithm is clever, no other modification on the network architecture or training method is reported. Generation of spectral libraries also followed previously developed methods by the authors. Maintaining quantitation accuracy and faithful FDR estimates are not surprising. I am not convinced there is enough novelty and contribution to the general community in this regard.

Second, the new algorithm has not been extensively tested and systematically compared with previous ones. For example, one would wish to see a comparison of number of precursors as a function of estimated FDR, for IM-DIA-NN vs openSWATH, similar to that in the original DIA-NN paper. Also, no further experimental datasets were provided or tested, this raises the concern of the reproducibility and robustness of this method. Another concern is, if, by the authors' estimate, better spectral library contributed 600-700 more proteins, and each of the IM modules about 400 proteins, then, what the observed difference between protein ids by IM-DIA-NN and openSWATH (5047-2936=2111) would have been mostly explained. This raises the question of what exactly is the role of deep neural network here and how much it really contributes. Most likely this is not a problem, and the Δ protein ids are not additive, but the close numbers still raise a concern and it's worth looking into.

Lastly, the paper is not written in a clear and concise way, with repetitive sentences and unnecessary quotes of too many numbers in the text, e.g. more than half of Fig. 2a is already describe by the text. On the other hand, the figures are not clear enough, e.g., to make effective comparisons, the protein id numbers from the original diaPASEF paper should be illustrated side-by-side as the new method.

Reviewer #2:

Remarks to the Author:

The manuscript submitted by Demichev et al. titled "Neural network-based processing of ion mobility data for deep high-throughput proteomics of low sample amounts" describes a method for the analysis of ion mobility mass spectrometry-based proteomics data. For this, the authors have modified the data processing in DIA-NN, a neural-network assisted data processing pipeline for DIA data, and combined this tool with the MSFragger pipeline. The authors highlight the application of their approach on a variety of publicly available data, particularly with the focus on samples with low input amounts. They show that their method increases the number of confident identifications from spectrum to protein level and produces accurate and consistent quantification estimates. The topic of low samples amounts in proteomics is of high interest. In addition, only rather recently support for trapped ion mobility (TIMS) data in proteomics is emerging.

Major issues:

- The changes necessary made by the authors seem incremental and neither new data or novel insight into existing data is provided in this manuscript. I was not able to find the git commit introducing the TIMS support. As far as I can tell from the manuscript, the biggest changes were made in DIA-NN. However, no detailed description is provided on what exactly was done to support this. The text describing the MSFragger-based library building is more extensive than those for DIA-NN. The vague descriptions in the results section of the manuscript e.g. "Among these putative peaks, different ones might be eventually used during subsequent targeted chromatogram extraction, depending on the reference ion mobility and fragment masses of a particular query precursor ion, thus maximizing the sensitivity." are not sufficient to judge or allow the re-implementation of the proposed changes.
- The FDR estimation of DIA-NN does not seem to be well calibrated (Figure S3; showing peptide or protein FDR?). While it is certainly less of a problem to provide conservative estimates, I wonder why this is the case. Particularly because the FDR calibration shown in the original publication of DIA-NN shows a better agreement and thus may suggest that the changes made here for diaPASEF data introduced this issue. Can the authors comment on this? In the original publication, the authors mentioned/cite previous work that showed that even slight changes to the way decoy peptides are generated may result in a strong deviation of the estimated and real FDR.

Minor issues:

- As far as I understand, no changes were made to the ensemble neural network, apart from adding TIMS information as additional input?
- I suggest reordering the subsections. First, I recommend moving the section on the gas-phase fragmentation into the supplement, as it does not add anything to the major storyline of the paper. Second, after identification, I recommend showing first the quantitative accuracy and subsequently show the CV. As a reader, I was wondering first how accurate this system is as otherwise CV values might simply show that the same error (incorrect identification/quantification) is made consistently across all replicates.
- The sentence "We report substantial gains of up to 72% in proteomic depth and quantification accuracy" implies that the quantification accuracy was improved by 72% as well. As far as I can tell, the improvement on quantification was not quantified. Second, to illustrate the improvement on identification, I recommend adding the original numbers to Figure 1a as this seem to be the major selling point of the method. On this note, please add an analysis on the overlap between the originally published precursors and peptides (and if possible also proteins) and the results obtained by the new workflow. The authors might further make the point of the benefits when using the TIMS dimension by showing some examples(s) of data where the previous workflow(s) failed to identify a particularly good scoring hit gained by MSFragger/DIA-NN.
- As far as I can tell, DIA-NN generates matching decoy library entries for each target entry by "mutating" the amino acid adjacent to the peptide termini. Due to the very strong correlation between CCS and m/z, altering the precursor mass of the target peptides, but retaining the original CCS values might lead to generating a feature with tells targets from decoys apart – because the m/z does not match the CCS values. Did the authors investigate alternative strategies for generating decoy? This potential issue might not be present right now as MSFragger does not score the deviation of the m/z to CCS and thus some incorrect targets will have non matching CCS values, however, alternative strategies for generating spectral libraries might inadvertently generate libraries where the difference in m/z to CCS can be picked up by DIA-NN.
- The clarity of Figure 1 (particularly panel ion mobility score) should be improved. It is not clear to me what the 1/K0 numbers indicate (start, end, apex) given that the green curve appear slightly shifted one would have expected this number to be different. I assume the traces are along the TIMS dimension? Similarly, it is not clear to me what the arrow relates here. Is the library entry 1.13? Why is one of the peaks detected in the earlier panel marked red? I suppose the colored lines represent the peaks colored in blue earlier? If so, one might color the peaks according to the lines in the ion mobility panel. Additionally, what does the dotted black line represent? And how is the outlier 1/K0 used?
- The clarity of Figure 2 should be improved. The authors state that in the legend for panel (a) that "Numbers of proteins detected in 1, 2 or all 3 replicates for each dataset [...] are shown with different color shades. I am only able to see 2 color shades. For panel (b), please indicate the 'n' of the violin distributions.
- Figure 3 misses a legend for the colors used and a description of the black horizontal line. The "--" on both y-axes seems to be placed by hand as it partially overlaps with the numbers.

Typos and other editing notes:

- „we speculated that further grains in dia-PASEF performance can be obtained with an” – I guess this should read “gains” and not “grains”
- The authors switch between different styles citations (i.e. Meier et al., 2020 should be [13])
- “(200 SPD (samples per day) Evosep One method)” – maybe the authors find a better way of introducing the abbreviation SPD to avoid the double brackets.

**Summary of the revisions made**

We thank the reviewers for their thoughtful comments. We have now clarified the descriptions
and added new data, extra benchmarks, comparisons and illustrations, as suggested. Below we
provide a detailed response to all questions and concerns raised.

We would like to highlight that our workflow is not just a combination of previously available tools,
and apologize if we created this impression in our first manuscript version, because we integrated
the new algorithms into the existing DIA-NN and FragPipe frameworks, to facilitate their ease of
use. A significant proportion of the performance gains are achieved through conceptually new
algorithms. The DIA ion mobility module itself took over a year to implement and test, and in the
revision we now highlight why we think it is crucial for enabling high sensitivity as well as excellent
performance on fast-gradient runs. We believe the impact of this manuscript is hence two-fold.
The concepts behind the new algorithms are of interest to the specialists in the field, as they bring
a new concept into DIA & TIMS proteomic data processing. Second, a broader impact is
generated by providing a powerful and easy to use software solution that enables high-quality
experiments, and that reaches significant improvements in proteomic depth, as demonstrated
below.

**The highlights of the revision are:**

1. Improved performance. For instance, we now quantify more proteins from 10ng than
previously (Meier et al, 2020) from 50ng, and more proteins using a 5.6-minute
chromatographic gradient than previously (Meier et al) with a 21-minute gradient (Figure
2a).
- 2. As requested by the Reviewers, we have conducted new experiments (i.e. added extra
benchmarks of the software recording our own data and also using additional public data),
as well as compared the performance to Spectronaut, the leading commercial dia-PASEF-
capable software. We report very significant gains in ID numbers, identification confidence
and data reliability.
- 3. We showcase the ultimate potential of dia-PASEF combined with state-of-the-art data
processing, by benchmarking the second-generation Bruker instrument, timsTOF Pro 2.
For example, we obtain 7400 proteins from 10ng HeLa, with a 93-minute 300nl/min
gradient. This is an increase in ID numbers of 92% compared to the state-of-the-art just a
36 year ago (Meier et al).

We also introduced two corrections: 1. In the first submission, we wrote that the Evosep 200 SPD
method uses a 4.8-minute gradient. We learned in the process that the runtime of this Evosep
method is in fact 5.6-minute for 200 SPD. This has now been corrected in the text. 2. Previously
we inadvertently reported larger numbers of proteins for OpenSWATH than OpenSWATH actually

identified. This was because the pyprophet reports deposited by Meier et al. also contained a
minor number of decoy proteins, whereas we previously assumed that these had been filtered
out. We have now corrected the numbers for OpenSWATH (e.g. previously we reported that
OpenSWATH identified 2936 proteins from Evosep 200SPD data, but after correction it is only
2902).

**Reviewer #1**

*1. In this manuscript, Demichev et al. present software algorithms and optimized spectral libraries*
*that facilitate processing of ion mobility information in diaPASEF data. The authors extended their*
*previously developed DIA-NN framework by adding a 2D peak picking algorithm and extracting*
*several IM related information as neural network inputs. The authors also prepared specific*
*spectral libraries suitable for use with diaPASEF data analysis. The authors benchmarked their*
*IM-DIA-NN algorithm on previously published datasets, and showed generally improved*
*performance over the original diaPASEF report, with a significant increase in protein identification*
*in certain cases (up to 72%). The authors also studied the quantitation accuracy, and looked into*
*effects of semi-tryptic peptides.*

*Increasing proteomics throughput and using lower sample amounts without compromising*
*proteomic depth is clearly one of the key directions for future proteomics, and this manuscript*
*reports significant gains in protein identification in certain small samples (similar number of*
*proteins ids from a 5-min gradient as the original 12-min gradient).*

We are glad that the reviewer appreciates very significant gains over state-of-the-art achieved by
our method. Here we would like to note that in the revision we present a significantly improved
workflow. At strict FDR settings, our workflow now yields more proteins in ~5-minutes (Evosep
200 SPD method) than the original workflow in ~20-minutes (Evosep 60 SPD) and detects more
proteins in 10ng of a HeLa extract than the original workflow from 50ng. In addition, using the
second-generation timsTOF Pro 2, we now further almost doubled the number of proteins
quantified from 10ng, to 7400.

*2. Nonetheless, I am unable to recommend this paper for publication in Nature Communications*
*based on the following grounds. First, the DIA-NN neural network framework has been developed*
*previous by the authors, and what's new here is the addition of a 2D peak picking algorithm (and*
*matching IM values). While this algorithm is clever, no other modification on the network*
*architecture or training method is reported. Generation of spectral libraries also followed*
*previously developed methods by the authors. Maintaining quantitation accuracy and faithful FDR*
*estimates are not surprising. I am not convinced there is enough novelty and contribution to the*
*general community in this regard.*

We apologize that the Reviewer came to the conclusion that the performance gains are explained
solely by the combination of existing software algorithms, which is not the case. Indeed, we
present a suite of algorithms that centre around a new 2D-peak-picking and quantification strategy

that benefits from neural network-based scoring. We have now significantly reworked the
description of the algorithms, and we specifically illustrate it (Figure 1) to highlight in the main text
why 2D-peak-picking is fundamentally different from profile-data-based approaches used
previously in the analysis of ion mobility proteomics data.

The main advantage of the new method is that it fully integrates each peak in the $1/K_0 \times m/z$
space, but at the same time minimizes integration of interferences. That is part of the reason why
we achieved a much better performance, compared to other software (OpenSWATH,
Spectronaut) that are extracting chromatograms directly from the profile data. Compared to our
new algorithm, in profile-extraction either much of the ion signal is lost due to narrow extraction
windows, or, as we believe is likely the case with previous methods, a large extraction window
leads to integration of extra interfering signals. The new ion mobility extraction strategy is thus
the first to be devoid of the drawbacks of profile-extraction approaches, which we believe is the
key to the performance advantage it demonstrates.

The reason we have integrated the new software in DIA-NN rather than generating a new
standalone software, is simply to accelerate the distribution, and to make it maximally easy to
use. We believe that the integration into a popular software suite is a strength of the study, rather
than a weakness, as it significantly improves the take up of the method and the impact of the
paper.

*3. Second, the new algorithm has not been extensively tested and systematically compared with*
*previous ones. For example, one would wish to see a comparison of number of precursors as a*
*function of estimated FDR, for IM-DIA-NN vs openSWATH, similar to that in the original DIA-NN*
*paper.*

In the first submission we replicated the benchmarks as they were deemed sufficient in the original
dia-PASEF paper by Meier, Mann et al, which presented the acquisition technology itself, plus
the first software (Meier et al, 2020). We further added an extra FDR-validation benchmark. That
being said, we agree with the Reviewer that additional benchmarks will strengthen the confidence
in our workflow. We have hence added additional benchmarks. As a result, our software is now
more comprehensively benchmarked than previous software presented for the processing of dia-
PASEF data.

First, we benchmarked our workflow against Spectronaut on data generated for a study entitled
'The Protein Landscape of Chronic Lymphocytic Leukemia' (Meier-Abt et al, 2021, PXD022216).
In this dataset, 50 cancer samples are analysed with 100-min 400nl/min gradient on a timsTOF
Pro using dia-PASEF. Compared to Spectronaut, we demonstrate a 72% gain on the precursor
level and 58% on the protein level (Figure 2c). To put this result into perspective, this performance
boost achieved by our DIA ion mobility module exceeds the advantage in proteomic depth
demonstrated by 95-minute gradient nanoflow runs over 5.6-minute gradient Evosep runs (Figure
2a).

Further, we also added benchmarks on data acquired with the second generation timsTOF
(timsTOF Pro 2) instrument in our own laboratory (Figure 2e). We performed a dilution series of
human cell extract proteomes. We report to our knowledge unprecedented performance numbers
in single-injections and low sample amounts, e.g. 8910 proteins quantified in 93-minute 300nl/min
gradient from 200ng, 7400 from 10ng or 3659 from 1ng. This corresponds to a 92% increase in
IDs over previously reported numbers from 10ng with the same 300nl/min flow rate (3863 from
10ng reported previously: Figure 2a, see also Meier et al, 2020). Indeed, to our knowledge this
revision is the first benchmark of the timsTOF Pro 2 running DIA acquisition, and we believe that
the unprecedented performance we report is an extra reason the manuscript will gain a lot of
traction.

Eventually, to dispel any potential concerns that the performance gains might be achieved through
an inflation of false discoveries, we included an additional benchmark in which FDR testing is now
performed against Spectronaut (Figure S2). Here, with longer gradients, we are demonstrating
roughly a two-fold improvement in precursor numbers. Further, we are demonstrating that our
workflow is able to analyse data with short chromatographic gradients, which is still a limitation of
alternative software that focuses on conventional long gradients (e.g. Spectronaut,
OpenSWATH). For instance, while our new workflow quantifies ~30000 precursors at 1%
experimentally validated FDR, as estimated with the two-species method in the respective
benchmark (5.6-min gradient; Figure S2), Spectronaut achieves ~4000-8000 precursors per run
using the same data at equal FDR (Figure S2).

Unfortunately, we tried running OpenSWATH with exactly the same settings as used by Meier et
al, but it kept exiting with an error "Requested ion mobility extraction but no ion mobility array
found". The raw file was converted to .mzML using the supplied script and was being loaded
correctly, with this error only occurring at the peak extraction step.

*4. Also, no further experimental datasets were provided or tested, this raises the concern of the*
*reproducibility and robustness of this method.*

To address this comment, the revision is now a mixture of public datasets and our own. Please
see our reply to the comment #3 above.

However, we would like to state that it was intentional to benchmark the software based on
publicly available datasets (i.e. as aforementioned, we replicated the benchmarks from the Meier
et al, 2020 study). We and others have done so also in our previous algorithm papers, as
benchmarking gives best confidence with the readers if conducted on data that is recorded
independently and is publicly available. Second, one avoids the danger that one over-optimizes
the software for the setup in our own lab, and allows others that want to replicate our results to
do so without any restriction. However, we agree that now doing both, benchmarking on own and
public data, gives even more confidence, and hence we added the above mentioned datasets.

*5. Another concern is, if, by the authors' estimate, better spectral library contributed 600-700 more*
*proteins, and each of the IM modules about 400 proteins, then, what the observed difference*

*between protein ids by IM-DIA-NN and openSWATH (5047-2936=2111) would have been mostly*
*explained. This raises the question of what exactly is the role of deep neural network here and*
*how much it really contributes. Most likely this is not a problem, and the Δ protein ids are not*
*additive, but the close numbers still raise a concern and it's worth looking into.*

We apologize if the presentation of our results created the impression that the gains are only due
to the improved spectral libraries. We reported, in the first submission, +702 proteins from a
FragPipe-based spectral library and +1062 from due to the new algorithms for ion mobility
processing, for a 5-minute gradient. Even subtracting these numbers together (and as reviewer
noted, the cumulative effect is likely less), we get $5047 - 702 - 1062 = 3283$, which is still a
substantial gain compared to the performance of OpenSWATH with its fully-enabled ion mobility
module.

*6. Lastly, the paper is not written in a clear and concise way, with repetitive sentences and*
*unnecessary quotes of too many numbers in the text, e.g. more than half of Fig. 2a is already*
*describe by the text. On the other hand, the figures are not clear enough, e.g., to make effective*
*comparisons, the protein id numbers from the original diaPASEF paper should be illustrated side-*
*by-side as the new method.*

We apologize, and have now significantly reworked the text and added the numbers achieved in
the original work using MaxQuant and OpenSWATH to the figures.

**Reviewer #2**

*1. The manuscript submitted by Demichev et al. titled "Neural network-based processing of ion*
*mobility data for deep high-throughput proteomics of low sample amounts" describes a method*
*for the analysis of ion mobility mass spectrometry-based proteomics data. For this, the authors*
*have modified the data processing in DIA-NN, a neural-network assisted data processing pipeline*
*for DIA data, and combined this tool with the MSFragger pipeline. The authors highlight the*
*application of their approach on a variety of publicly available data, particularly with the focus on*
*samples with low input amounts. They show that their method increases the number of confident*
*identifications from spectrum to protein level and produces accurate and consistent quantification*
*estimates. The topic of low samples amounts in proteomics is of high interest. In addition, only*
*rather recently support for trapped ion mobility (TIMS) data in proteomics is emerging.*

We thank the reviewer for appreciating the significance of our advances in the field of high-
sensitivity proteomics.

*2. Major issues:*

*- The changes necessary made by the authors seem incremental and neither new data or novel*
*insight into existing data is provided in this manuscript. I was not able to find the git commit*
*introducing the TIMS support. As far as I can tell from the manuscript, the biggest changes were*
*made in DIA-NN. However, no detailed description is provided on what exactly was done to*

*support this. The text describing the MSFragger-based library building is more extensive than*
*those for DIA-NN. The vague descriptions in the results section of the manuscript e.g. “Among*
*these putative peaks, different ones might be eventually used during subsequent targeted*
*chromatogram extraction, depending on the reference ion mobility and fragment masses of a*
*particular query precursor ion, thus maximizing the sensitivity.” are not sufficient to judge or allow*
*the re-implementation of the proposed changes.*

This comment is similar to comment 2 of Reviewer #1, and we apologize, and believe this
impression was caused by us not highlighting sufficiently that the performance gains are mostly
down to new algorithms, and the fact that we have incorporated the new software in DIA-NN and
FragPipe rather than creating a new standalone software.

As aforementioned (please see our detailed answer to Reviewer #1, comment 2, pages 1-2, lines
82-104), we have added an overview Figure that explains the new developments. Further, we
believe that the incorporation of our new algorithms into popular software pipelines (DIA-NN and
FragPipe) is a strength rather than a weakness, as it substantially facilitates traction and
distribution of our new developments.

*3. The FDR estimation of DIA-NN does not seem to be well calibrated (Figure S3; showing peptide*
*or protein FDR?). While it is certainly less of a problem to provide conservative estimates, I*
*wonder why this is the case. Particularly because the FDR calibration shown in the original*
*publication of DIA-NN shows a better agreement and thus may suggest that the changes made*
*here for diaPASEF data introduced this issue. Can the authors comment on this? In the original*
*publication, the authors mentioned/cite previous work that showed that even slight changes to the*
*way decoy peptides are generated may result in a strong deviation of the estimated and real FDR.*

In DIA-NN, similar to previous DIA software, FDR is estimated with a target-decoy method and is
based on the assumption that decoy-spectrum matches are a good model for false target-
spectrum matches. Fluctuations in FDR estimation accuracy depending on the dataset and
search settings are thus inevitable: no model is 100% perfect. We don't apply any 'artificial'
adjustment factors to FDR estimation, as this might be potentially dangerous and can lead to too
optimistic FDR being reported. The algorithms of DIA-NN with respect to FDR estimation and the
decoy generation scheme (the amino acid mutation pattern) are presented in the original DIA-NN
paper (Demichev et al, 2020) and in this work have proven applicable in the analysis of ion mobility
data. We would therefore explain the more conservative character of the estimates obtained here
by the use of π_0 correction ('prior probability of incorrect identification', also known as PIT -
Percentage of Incorrect Targets) when calculating FDR with the two-species method: in the
original DIA-NN paper π_0 correction was not used. So the performance demonstrated in this FDR-
validating benchmark is about what we would have expected based on the previous data. Both
precursor and protein FDR validation is shown on Figure S2 in separate plots.

We have now added Spectronaut's results to the FDR benchmark (Figure S2). Not only does our
new TIMS software produce a gain in ID numbers, but it also provides reliable FDR estimates
(especially in the 0%-1% region: the new software is more accurate than Spectronaut in this

region for all runs considered). As aforementioned, our software is in particular advantageous in
the analysis of fast gradients (5.6-minute here), which are one of the highlights of our work.

*4. Minor issues:*

*- As far as I understand, no changes were made to the ensemble neural network, apart from*
*adding TIMS information as additional input?*

It is correct, when neural networks are concerned, that our work builds on the core algorithms that
have been introduced with DIA-NN (Demichev et al 2020). However, the performance as
presented in this paper, is not achievable with the original software, lacking the new algorithms.

The TIMS module based on 2D-peak-picking, as presented here, could not easily be implemented
in other existing software frameworks. For example, it would require a complete redesign of
OpenSWATH to pair it with our TIMS module. The TIMS module is designed to be integrated with
a neural network classifier, and all scores calculated for a peak group, including specifically based
on the ion mobility dimension of the data, are processed by the neural networks. Thus, it's the
neural network processing which allows to fully take benefit from the ion mobility dimension. Minor
changes have been introduced to the architecture of the networks and the training procedure.
However the peak group scoring procedure, used to generate the input for neural networks, was
significantly refactored to differentially score fragment ions depending on the respective
deviations in ion mobility dimension.

As we described previously (Demichev et al, 2020), the training of neural networks is conducted
on a high number of scores for each peak group (over 70 were introduced with DIA-NN in 2020,
and herein we train with over 100 scores, to accommodate the new requirements of TIMS, and to
achieve further performance gains). These scores can only be aggregated effectively in a single
score using neural networks: with this dimensionality and with complex non-Gaussian scores a
linear method, like LDA, would typically become quite unstable and will produce aberrant high
discriminant scores for peak groups for which one or several scores out of 100+ are outliers. We
believe this effect with linear classifiers makes it difficult to achieve confident identification, while
neural networks seemingly don't exhibit this. For instance, retention time and ion mobility
differences are particularly difficult for the linear classifier. For example, we previously had to use
'square root of the absolute difference between measured and predicted retention times' as a RT
score (Demichev et al, 2020). The square root was applied to obtain better robustness for the
linear classifier, but even with such scaling this score still had to be disabled during the first
calibration stages of the search to avoid outlier effects. Neural networks don't seem to have such
problems. This makes them particularly advantageous when handling ion mobility data too.

*5. I suggest reordering the subsections. First, I recommend moving the section on the gas-phase*
*fragmentation into the supplement, as it does not add anything to the major storyline of the paper.*

We thank the reviewer for the suggestion and have removed the gas-phase fragmentation
discussion as it is of little relevance to the message of this work.

*6. Second, after identification, I recommend showing first the quantitative accuracy and*
*subsequently show the CV. As a reader, I was wondering first how accurate this system is as*
*otherwise CV values might simply show that the same error (incorrect identification/quantification)*
*is made consistently across all replicates.*

We agree with the reviewer that accuracy is at least as important as precision, and ideally should
be a major benchmark metric used in every methodological development, that's why we also
benchmarked accuracy of our method (Figure 3). Still, the adoption of accuracy benchmarks in
the field is rather slow, and most readers still expect to see primarily the ID numbers and the CV
values. We would therefore prefer to retain the CVs figure (Figure 2b) directly after the ID numbers
figure (Figure 2a), and also because it relates to the same datasets, whereas accuracy is
benchmarked on a different dataset. We however now highlight it in the main text that accuracy
is a highly important metric on par with CV values.

*7. The sentence "We report substantial gains of up to 72% in proteomic depth and quantification*
*accuracy" implies that the quantification accuracy was improved by 72% as well. As far as I can*
*tell, the improvement on quantification was not quantified.*

We thank the reviewer for spotting this and have now rephrased that sentence, to avoid confusion.
Indeed, it is difficult to quantify the accuracy improvement in a single metric. We have hence
decided to illustrate the accuracy improvement through visualisation only (Figure 3).

*8. Second, to illustrate the improvement on identification, I recommend adding the original*
*numbers to Figure 1a as this seem to be the major selling point of the method.*

We agree with the Reviewer, and have now added original numbers from MaxQuant +
OpenSWATH pipeline to the figures.

*9. On this note, please add an analysis on the overlap between the originally published precursors*
*and peptides (and if possible also proteins) and the results obtained by the new workflow. The*
*authors might further make the point of the benefits when using the TIMS dimension by showing*
*some examples(s) of data where the previous workflow(s) failed to identify a particularly good*
*scoring hit gained by MSFragger/DIA-NN.*

Thank you for this suggestion. We have now indicated on Figure 2a (with dashed lines) the
numbers of proteins jointly detected by our and the original workflows. We see that almost all
proteins detected originally are also detected by our workflow, but our workflow identifies and
quantifies large numbers of extra proteins. We have provided histograms for the intensity
distributions of shared/unique precursors and proteins in Figure S3. As suggested, we considered
also adding some chromatogram visualizations, but decided not to for two reasons: (1) it would
be difficult for us to figure out why OpenSWATH did not detect a particular peptide - might be
something as simple as the peptide not satisfying some tolerance threshold - so the visualization
is unlikely to be highly informative, and (2) we still lack easy-to-use visualization software for
PSMs on dia-PASEF data.

*10. As far as I can tell, DIA-NN generates matching decoy library entries for each target entry by*
*“mutating” the amino acid adjacent to the peptide termini. Due to the very strong correlation*
*between CCS and m/z, altering the precursor mass of the target peptides, but retaining the original*
*CCS values might lead to generating a feature with tells targets from decoys apart – because the*
*m/z does not match the CCS values. Did the authors investigate alternative strategies for*
*generating decoy? This potential issue might not be present right now as MSFragger does not*
*score the deviation of the m/z to CCS and thus some incorrect targets will have non matching*
*CCS values, however, alternative strategies for generating spectral libraries might inadvertently*
*generate libraries where the difference in m/z to CCS can be picked up by DIA-NN.*

We follow the same decoy generation procedure as we have described previously (Demichev et
al, 2020), as this strategy proved to be also suitable for the analysis of the ion mobility data.
Specifically, only fragment masses are changed in the decoy in comparison to the respective
target. The precursor m/z, relative fragment intensities, retention time and now the ion mobility -
all these remain exactly the same in the decoy as in the target. This is done specifically to exclude
the situation which the Reviewer brings up here. Before the original DIA-NN version (1.6) was
published, we indeed investigated alternative strategies for decoy generation, like also changing
the precursor m/z and RT values, different amino acid mutation patterns or different variations of
reverse/pseudo-reverse decoys. The ‘mutated decoys’ algorithm published in the original DIA-NN
paper and also used here is thus highly optimised.

*11. The clarity of Figure 1 (particularly panel ion mobility score) should be improved. It is not clear*
*to me what the 1/K0 numbers indicate (start, end, apex) given that the green curve appear slightly*
*shifted one would have expected this number to be different. I assume the traces are along the*
*TIMS dimension?*

We thank the reviewer for spotting this. Indeed, the green curve on the schematic is supposed to
indicate a fragment affected by slight interference. Indeed, it would be natural then to alter the
1/K0 value (apex) slightly - we have now adjusted and clarified the figure. Chromatographic
elution traces are shown; we have now indicated this in the figure legend.

*12. Similarly, it is not clear to me what the arrow relates here. Is the library entry 1.13? Why is*
*one of the peaks detected in the earlier panel marked red? I suppose the colored lines represent*
*the peaks colored in blue earlier? If so, one might color the peaks according to the lines in the ion*
*mobility panel. Additionally, what does the dotted black line represent? And how is the outlier 1/K0*
*used?*

We apologise for the confusion. We have now rewritten the description and hope that it is a lot
more clear now. 1.13 is indeed the library entry. The red denotes peaks rejected due to IM values
outside of the IM window (these peaks will not be considered during chromatogram extraction),
blue - accepted peaks, black - peaks rejected due to masses outside the m/z window. Dotted lines
represent expected masses & IM values. The outlier affects the calculation of scores which are

ultimately weighted by the neural networks, to produce a single composite score. We have now
added all this to the figure legend.

*13. The clarity of Figure 2 should be improved. The authors state that in the legend for panel (a)*
*that “Numbers of proteins detected in 1, 2 or all 3 replicates for each dataset [...] are shown with*
*different color shades. I am only able to see 2 color shades. For panel (b), please indicate the ‘n’*
*of the violin distributions.*

We have now enhanced the figures and indicated the numbers for violins.

*14. Figure 3 misses a legend for the colors used and a description of the black horizontal line.*
*The “-“ on both y-axes seems to be placed by hand as it partially overlaps with the numbers.*

We have now added the legend and indicated the meaning of the horizontal line. There was no
manual editing of the figures involved; this is how the plot was produced by the R script (included
in the submission). We have now corrected this manually.

*15. Typos and other editing notes:*

*- „we speculated that further grains in dia-PASEF performance can be obtained with an” – I guess*
*this should read “gains” and not “grains”*

*- The authors switch between different styles citations (i.e. Meier et al., 2020 should be [13])*

*- “(200 SPD (samples per day) Evosep One method)” – maybe the authors find a better way of*
*introducing the abbreviation SPD to avoid the double brackets.*

We thank the Reviewer for spotting these, and have corrected the manuscript.

**References**

Demichev, V., Messner, C. B., Vernardis, S. I., Lilley, K. S. & Ralser, M. DIA-NN: neural networks
and interference correction enable deep proteome coverage in high throughput. *Nat. Methods* **17**,
41–44 (2020). <https://www.nature.com/articles/s41592-019-0638-x>

Meier-Abt, F. et al. The Protein Landscape of Chronic Lymphocytic Leukemia (CLL). *Blood* (2021)
doi:10.1182/blood.2020009741. <https://doi.org/10.1182/blood.2020009741>

Meier, F. et al. diaPASEF: parallel accumulation–serial fragmentation combined with data-
independent acquisition. *Nat. Methods* **17**, 1229–1236 (2020).

<https://www.nature.com/articles/s41592-020-00998-0>

Reviewers' Comments:

Reviewer #1:

Remarks to the Author:

In the revised submission, the authors have replied to my previous comments and addressed some concerns. In particular, the authors further tested their software on sets of published and new experimental data, adding support to their algorithms' performance. The authors also showcased better performance (compared to previous version) in protein identification from small sample, using the newer instrument (timsTOF Pro 2). The authors also reorganized figures and text for better clarity to the readers, including a new figure panel to better illustrate the benefits of the 2D peak picking algorithm.

However, there is still concern of incremental changes that the authors have not completely addressed.

First, the authors seemed to be not careful in replying to comments. In my previous comments (and Reviewer #2 essentially asked the same question), I mentioned "what's new here is the addition of a 2D peak picking algorithm"; the authors replied, "We apologize the reviewer came to the conclusion that the performance gains are explained solely by combination of existing software algorithms", and "Indeed, we present a suite of algorithms that centre around a new 2D-peak-picking and quantification strategy". Similarly, reviewer #2's comments regarding "detailed description .. on what exactly was done" and "vague description" were not addressed. Instead, the authors simply removed two "vague" sentences from the text.

Second, in the replies as well as the revised manuscript, the authors stressed on the importance of using the 2D peak picking algorithm (as compared to a profile-extraction method). On the other hand, according to Fig. S1, "IM peak picking" seem to account for only minor increments (+250) in protein identification, the rest came from IM scoring and IM window (and what do these different "modules" exactly refer to, in the text description of the IM analysis algorithm?) It would be helpful if the authors could provide a direct comparison of 2D peak picking vs profile-extraction – if this is indeed the most significant part of the algorithm.

I would be willing to recommend publication for the manuscript if my above comments can be well addressed by the authors.

Reviewer #2:

Remarks to the Author:

The authors have added new analysis, datasets and benchmarks and are able to show impressive numbers on existing and new datasets when using DIA-NN. The authors have addressed some of my concerns, however, some remain open:

Major issue:

- Precisely because the authors claim that "the performance gains are mostly down to new algorithms", I asked for a detailed description of the changes. The authors did not address this. Neither the source code nor a written description is provided to allow the independent reproduction of the results. Without any details, I have a hard time judging whether the statement "The TIMS module is designed to be integrated with a neural network classifier [...]" is of relevance. As far as I can see, the TIMS module generates scores which could have been used for e.g. Percolator the same way as it was used for DIA-NN. In addition, the authors write (in the rebuttal) that "minor changes have been introduced to the architecture of the networks and the training procedure" and that "[...] peak group scoring procedure [...] was significantly refactored [...]", neither of which are described in the manuscript, particularly the latter. In fact, the manuscript does not even state that any of these changes were made. The claim that they "present a computational strategy [...] including novel algorithms" (in the rebuttal referred to as "a suite of algorithms") is not met.
- The remaining open issue of a potential lack of novelty does not originate from the fact that the IM support was implemented in DIA-NN directly. This is highly appreciated and welcome. The authors write in the rebuttal that "we believe [the new ion mobility extraction strategy is] the key to the performance advantage it demonstrates". Without supporting data, I am unable to simply

believe that the key performance advantages originate from the new (2D) ion mobility extraction approach. The approach is not compared directly to any other approach (e.g. using XIC extraction rather than peak picking in DIA-NN) and the integration of IM as an additional input to the neural network appears incremental. The gains reported are largely the result of DIA-NN (without IM support) as shown for the Evosep 200/100/60 SPD experiments (Figure 2 and S1). The gains introduced by IM-support appear to be in the range of ~13-25% (Figure S1, relative change in comparison to DIA-NN without IM peak picking) of which the majority seems to originate from using the IM dimension for feature extraction (without neural network-based processing). The last increase, the "Neural network-based processing" aspect of the title only adds 4-6% of the final (mean) results. One may even argue that the neural network is not used for IM data processing – only in the very last step during FDR estimation – and thus the title might be a bit misleading. I recommend reporting the increase in IDs separately, at least the number with and without IM support, as those are the core contribution of this paper.

Minor issue:

- Only minimal experimental details for the data underlying Figure 2e is provided.
- The observed gain of 92% (data from Figure 2e) should be reported separately for the change in instrumentation (e.g. re-analyzing the timsTOF Pro 2 with the software used in ref 14) and software (e.g. re-analyzing the timsTOF Pro data with the new DIA-NN software).
- The color coding for proteins identified in 1, 2 and 3 replicates is still not visible (Figure 1a, d; Figure S1), particularly for blue, green and red. Only after highlighting the figures, the respective areas became barely visible.
- Figure 1c lacks y-axis labels.
- Figure 3 was modified. Why does the new manuscript show more outliers in comparison to the previous version? I was not able to spot the difference leading to this.

Summary

We thank the reviewers for constructive comments. We are glad that we were able to successfully address all technical concerns and that both reviewers now agree that the performance advantage over existing software strategies (an increase of up to 83% protein identifications at experimentally validated FDR), that is shown by our workflow, has been properly benchmarked and validated. We noticed however that both reviewers still expressed concern about whether the algorithms that comprise the TIMS module of DIA-NN are truly novel. We believe that we might have inadvertently triggered this impression ourselves, by not giving enough details how the new algorithms, in particular the 2D-peak-picking, work. In revision, we address their concerns, in the way suggested by the reviewers, specifically:

- We now implemented extra functionality in DIA-NN, which allows us to directly benchmark our novel 2D-peak-picking algorithm against the method of chromatogram extraction from the profile data described previously by Meier et al (2020), on the same software platform. The benchmark shows a significant advantage of our approach. We further provide the full detailed description of the 2D-peak-picking algorithm and the corresponding chromatogram extraction procedure.

In addition:

- We have further been able to improve some performance characteristics of DIA-NN. We also noticed that in an extra benchmark against Spectronaut on a public leukemia dataset, which we added in revision following reviewers' request for more thorough benchmarking, we inadvertently used a full uniprot human database, as opposed to a curated-only database used by the authors of the dataset. To allow for a more fair comparison, we have now likewise analyzed that dataset using a curated database, which reduced slightly the protein (but not precursor) numbers, without affecting the general conclusions.
- As requested by the reviewers, we now likewise show that the neural network classifier implemented in DIA-NN also provides a significant advantage for the processing of dia-PASEF data.
- We have reworked the figures, improving the presentation of the benchmarks and addressing the readability concerns raised by Reviewer 2.

Reviewer #1

In the revised submission, the authors have replied to my previous comments and addressed some concerns. In particular, the authors further tested their software on sets of published and new experimental data, adding support to their algorithms' performance. The authors also showcased better performance (compared to previous version) in protein identification from small sample, using the newer instrument (timsTOF Pro 2). The authors also reorganized figures and text for better clarity to the readers, including a new figure panel to better illustrate the benefits of the 2D peak picking algorithm.

1.1. We thank the reviewer for appreciating the improvements implemented in the revision.

However, there is still concern of incremental changes that the authors have not completely addressed. First, the authors seemed to be not careful in replying to comments. In my previous comments (and Reviewer #2 essentially asked the same question), I mentioned "what's new here is the addition of a 2D peak picking algorithm"; the authors replied, "We apologize the reviewer came to the conclusion that the performance gains are explained solely by combination of existing software algorithms", and "Indeed, we present a suite of algorithms that centre around a new 2D-peak-picking and quantification strategy". Similarly, reviewer #2's comments regarding "detailed description .. on what exactly was done" and "vague description" were not addressed. Instead, the authors simply removed two "vague" sentences from the text.

1.2. We apologize for our confusion in answering the comments. We have substantially worked on the revision, and give a far more detailed description of the 2D-peak-picking algorithm (Methods).

Indeed, however, the impact of our paper is not only explained by the conceptual novelty of the algorithms, but also by the performance gains we provide for the community. We believe that one strongest indicator that people really profit from our work is that the inventors of dia-PASEF themselves, the Mann lab, have recently been publishing dia-PASEF results using DIA-NN, and Bruker, the company that produces the timsTOFs, has re-written DIA-NN and is now shipping this software with the instrument.

- We implemented a pipeline that outperforms data processing as published in the timsTOF benchmark dataset (Meier et al, 2020, Nature Methods) by up to 83% in terms of protein numbers. Our pipeline has further been recently tested independently, and has been used to enable a label-free single-cell proteomics dia-PASEF workflow <https://doi.org/10.15252/msb.202110798>.
- The above is achieved by
 - A novel 2D-peak-picking algorithm at the core of the new TIMS module in DIA-NN. Idea-wise this is the key novelty to the field, as it changes the paradigm of how ion mobility-resolved data should be processed. The

benchmark requested by the reviewers is now provided (Supp. Figure S2, also please see the reply 1.3 below). A full detailed description of the algorithm is now also provided in Methods.

- DIA-NN uses ion mobility information to score PSMs. The way this is done is DIA-NN-specific, and we believe is unlikely to be of the same interest to the software developers as the 2D-peak-picking strategy, and thus is less significant novelty-wise. Still, some aspects of the way DIA-NN is doing the scoring are novel and unique: for example, it scores lower fragment ions the signal for which is an ‘outlier’ in terms of ion mobility (Figure 1); it also summarizes the ion mobility information using its neural networks classifier, which is unique to DIA-NN (no other DIA software uses it by default, to our knowledge) and has its advantages, as we described in the previous revision response.
- Integrating and benchmarking FragPipe and DIA-NN. This part is purely technical, however it still must be communicated, to allow scientists to make informed decisions on which workflow to use for dia-PASEF data processing.

Multiple other aspects of DIA-NN have been continuously improving since our initial release of the software in 2018, however they are not specific to dia-PASEF and we consider them beyond the scope of this work. To answer the reviewer’s question, we don’t claim any dia-PASEF-specific improvements to the DIA-NN’s neural network classifier. Nevertheless, the performance shown by our pipeline is significantly enabled by the use of the neural network classifier, as we now show in Supp. Figure S2. As neural networks are key to the reliable peptide identification shown by DIA-NN, as we demonstrated previously (<https://www.nature.com/articles/s41592-019-0638-x>), and are significantly responsible for the ultimate performance of our FragPipe + DIA-NN pipeline, we believe the mention of neural networks in the title is warranted.

Second, in the replies as well as the revised manuscript, the authors stressed on the importance of using the 2D peak picking algorithm (as compared to a profile-extraction method). On the other hand, according to Fig. S1, “IM peak picking” seem to account for only minor increments (+250) in protein identification, the rest came from IM scoring and IM window (and what do these different “modules” exactly refer to, in the text description of the IM analysis algorithm?) It would be helpful if the authors could provide a direct comparison of 2D peak picking vs profile-extraction – if this is indeed the most significant part of the algorithm.

1.3. as requested by the Reviewer, in order to be able to directly compare the chromatogram extraction algorithms, we studied the source code of OpenSWATH and have now implemented a module in DIA-NN which allows for direct extraction of chromatograms from the profile dia-PASEF data, replicating the algorithm used by OpenSWATH. This allowed for a direct benchmark of the 2D-peak-picking vs profile extraction (Supp. Figure S2). The results show that the shorter the chromatographic gradient (and hence lower the peak capacity and higher

the impact of interferences), the greater the advantage of the 2D-peak-picking method. The 2D-peak-picking method reaches a 27% higher ID rate at 200 SPD (for comparison, the difference between the 60 SPD and the three times faster 200 SPD is just 30%), as well as provides better data completeness.

As for the drop in performance DIA-NN experiences when 2D-peak-picking is disabled (-322 proteins for 200 SPD in the revision, Supp. Figure S1), this figure does not reflect the true impact of 2D-peak-picking. Indeed, we test disabling 2D-peak-picking while everything else related to ion mobility (such as using IM extraction window and scoring IM information for each PSM) is also disabled. This is a necessity, as it's technically impossible to turn off peak picking but keep the IM scoring. Thus, the 'effect' of peak picking in Figure S1 seems to be a lot less than it actually is in fully functional DIA-NN, as all other components of DIA-NN's TIMS module are not operating here. We thank the reviewer for suggesting a direct test against profile extraction, as profile extraction is indeed compatible with all downstream IM-specific scoring steps used by DIA-NN. In comparison to profile extraction, the advantage of 2D-peak-picking now reaches 1145 proteins for 200 SPD, clearly demonstrating that this is the better approach.

To clarify the Supp. Figure S1, we now introduce the concept of an IM window in the main text algorithm description, as well as explain the meaning of 'IM scoring' in the Supp. Figure S1 legend.

Reviewer #2

The authors have added new analysis, datasets and benchmarks and are able to show impressive numbers on existing and new datasets when using DIA-NN. The authors have addressed some of my concerns, however, some remain open:

We thank the Reviewer for appreciating the amount of work we invested in improving the software and the manuscript.

Major issue:

- Precisely because the authors claim that "the performance gains are mostly down to new algorithms", I asked for a detailed description of the changes. The authors did not address this. Neither the source code nor a written description is provided to allow the independent reproduction of the results. Without any details, I have a hard time judging whether the statement "The TIMS module is designed to be integrated with a neural network classifier [...]" is of relevance. As far as I can see, the TIMS module generates scores which could have been used for e.g. Percolator the same way as it was used for DIA-NN. In addition, the authors write (in the rebuttal) that "minor changes have been introduced to the architecture of the networks and the training procedure" and that "[...] peak group scoring procedure [...]"

significantly refactored [...]”, neither of which are described in the manuscript, particularly the latter. In fact, the manuscript does not even state

that any of these changes were made. The claim that they “present a computational strategy [...] including novel algorithms” (in the rebuttal referred to as “a suite of algorithms”) is not met.

2.1. We apologize for the confusion, we have now clarified explicitly what we claim as novel specifically in this work, please see our comment **1.2** in response to Reviewer 1. While the main impact and novelty of our work is the ultimate performance of our workflow, when it comes to algorithms, the main novelty is provided by the 2D-peak-picking method, for which a full detailed description is now provided in Methods. We have also now expanded our software to include both alternative peak picking algorithms, and can therefore now provide a benchmark of 2D-peak-picking against profile extraction, showing the significant performance gain enabled by the novel 2D-peak-picking (Supp. Figure S2, please also see our comment **1.3** in response to Reviewer 1). We also show the significant performance boost achieved by using neural networks in DIA-NN, as opposed to a simple linear classifier (Supp. Figure S2).

- The remaining open issue of a potential lack of novelty does not originate from the fact that the IM support was implemented in DIA-NN directly. This is highly appreciated and welcome. The authors write in the rebuttal that “we believe [the new ion mobility extraction strategy is] the key to the performance advantage it demonstrates”. Without supporting data, I am unable to simply believe that the key performance advantages originate from the new (2D) ion mobility extraction approach. The approach is not compared directly to any other approach (e.g. using XIC extraction rather than peak picking in DIA-NN) and the integration of IM as an additional input to the neural network appears incremental. The gains reported are largely the result of DIA-NN (without IM support) as shown for the Evosep 200/100/60 SPD experiments (Figure 2 and S1). The gains introduced by IM-support appear to be in the range of ~13-25% (Figure S1, relative change in comparison to DIA-NN without IM peak

picking) of which the majority seems to originate from using the IM dimension for feature extraction (without neural network-based processing). The last increase, the “Neural network-based processing” aspect of the title only adds 4-6% of the final (mean) results. One may even argue that the neural network is not used for IM data processing – only in the very last step during FDR estimation – and thus the title might be a bit misleading. I recommend reporting the increase in IDs separately, at least the number with and without IM support, as those are the core contribution of this paper.

2.2. We now prove with a benchmark the significant advantage of our novel 2D-peak-picking algorithm in comparison to the previously described (Meier et al, 2020) chromatogram extraction from profile data, please see Supp. Figure S2 and our comment **1.3** in response to Reviewer 1. We would like to highlight that our pipeline has further been tested independently, and has recently been used to enable a label-free single-cell proteomics dia-PASEF workflow <https://doi.org/10.15252/msb.202110798>.

As for the seemingly low drop in performance DIA-NN experiences when 2D-peak-picking is disabled (-322 proteins for 200 SPD in the revision, Supp. Figure S1), this figure does not reflect the true impact of 2D-peak-picking. Indeed, we test disabling 2D-peak-picking while everything else related to ion mobility (such as using IM extraction window and scoring IM information for each PSM) is also disabled. This is a necessity, as it's technically impossible to turn off peak picking but keep the IM scoring. Thus the 'effect' of peak picking in Figure S1 seems to be a lot less than it actually is in fully functional DIA-NN, as all other components of DIA-NN's TIMS module are not operating here. With a direct benchmark against profile extraction (Supp. Figure S2) the true advantage of 2D-peak-picking is now clear. This shows, that although the core algorithms of DIA-NN of course significantly contribute to the ultimate performance of our workflow, the effect of the totally novel 2D-peak-picking is also very substantial.

We further quantify the total effect of neural networks (Supp. Figure S2), which proves to be quite significant (+786 proteins for 200 SPD). We believe it justifies mentioning neural networks in the title.

Minor issue:

- Only minimal experimental details for the data underlying Figure 2e is provided.

We thank the reviewer for spotting this; we have now added the missing details (Methods), including the information on the pre-column, the ion source, as well as illustrated the dia-PASEF acquisition schemes used (Supp. Figure S5).

- The observed gain of 92% (data from Figure 2e) should be reported separately for the change in instrumentation (e.g. re-analyzing the timsTOF Pro 2 with the software used in ref 14) and software (e.g. re-analyzing the timsTOF Pro data with the new DIA-NN software).

Unfortunately, we cannot benchmark OpenSWATH on this dataset. As indicated in the response to the previous revision, we could not manage to process dia-PASEF data with OpenSWATH on our computers. However we would like to highlight that this benchmark was added (along with two benchmarks comparing our software against Spectronaut) in the previous revision upon the request of Reviewer 1 for more thorough benchmarking. We don't compare directly between the two instruments or software tools in this benchmark, but just see this benchmark as informative of the performance of our workflow.

- The color coding for proteins identified in 1, 2 and 3 replicates is still not visible (Figure 1a, d; Figure S1), particularly for blue, green and red. Only after highlighting the figures, the respective areas became barely visible.

We thank the Reviewer for pointing this out, now the color scheme has been inverted to guarantee high contrast.

- Figure 1c lacks y-axis labels.

Precursor and protein numbers are labeled as such, while the y-axis represents the counts as in a typical histogram. We have now clarified this in the figure legend.

- Figure 3 was modified. Why does the new manuscript show more outliers in comparison to the previous version? I was not able to spot the difference leading to this.

We believe this represents a ‘random’ effect, caused by the minimal changes in algorithms. These affect quantities of some proteins, potentially changing the numbers of outliers. In the present revision the number of outliers has been reduced, however, to our understanding, we again have not changed algorithms in any way which would affect quantification accuracy in general.

Reviewers' Comments:

Reviewer #1:

Remarks to the Author:

In the newly revised submission, the authors have now addressed my main concerns on the manuscript. Specifically, the authors added a detailed description of the new 2D peak picking algorithm in Methods, and perform direct comparison between 2D peak picking and profile extraction in Fig. S2, as well as other clarifications and better graphics. From the description, the algorithm itself is not a complicated one, essentially a 2D moving window average followed by exclusion of overlapping maxima. However, the performance of the algorithm is now solidly benchmarked, and shows significant improvement in peptide and protein ID over previous methods. I agree with the authors that the impact of the manuscript lies also in its significant performance gain over previous reports, in particular over shorter gradients, and the readily useful software packages for the general community. I believe the manuscript is overall suitable for publication at Nature Communications, after some minor corrections.

Minor issue:

- Given the main novelty of the work is in the introduction of a 2D peak picking algorithm, the authors should clarify this sentence in the abstract, and maybe a few other places: "We present software algorithms and the generation of optimized spectral libraries for neural network-based processing of ion mobility data acquired with dia-PASEF ... ", such that it is clear to the reader where the novelty and contribution of the current manuscript is - rather than leaving a vague impression that this work is the first one that uses neural network-based method for dia-PASEF data analysis.
- The 2D peak picking algorithm involves quite a few manual parameter settings, e.g. for the maximum ion mobility tolerance and m/z tolerance bounds, which are at the heart of the improvement performance claimed here. It will be helpful to the readers if the authors can add a short sentence describing how these parameters were chosen or tuned during development of the software.

Reviewer #2:

Remarks to the Author:

The authors have added new analysis, benchmarks, figures and modified the main manuscript to accommodate the main aspects of the requested changes. However, some details still remain open:

The authors mentioned in their earlier rebuttal that some changes were made on the architecture of the NN-based classifier and that the peak-group scoring was refactored. None of these changes are mentioned or described in this manuscript. I welcome that DIA-NN is freely available, however, the scientific community is not able to re-implement the advances discovered and proposed by DIA-NN in their own workflows without either 1) the source code or 2) a description of the changes. At least the changes on the neural architecture need to be described.

I remain skeptical that the title is an appropriate description of the main contribution of the manuscript. The neural network-based classifier is one of the key components of the published DIA-NN workflow and is arguably not used for processing the (raw) IM data. It "merely" aggregates information of various scores/metrics and as such is not aware that some scores represent IM information. The aspect that an NN-classifier increases IDs was highlighted in the original publication. As the authors describe, the main (algorithmic) contribution of this manuscript is the 2D-peak-picking, which is disconnected from the NN-based classifier. While the combination may still result in better numbers, the aspect of the NN-classifier is not novel anymore. I leave the final decision to the editor, but a title like "Integration of ion mobility data into neural network-based classifier for deep proteomics of low sample amounts" seems more appropriate and would still contain the relevant buzz words.

Reviewer #1

In the newly revised submission, the authors have now addressed my main concerns on the manuscript. Specifically, the authors added a detailed description of the new 2D peak picking algorithm in Methods, and perform direct comparison between 2D peak picking and profile extraction in Fig. S2, as well as other clarifications and better graphics. From the description, the algorithm itself is not a complicated one, essentially a 2D moving window average followed by exclusion of overlapping maxima. However, the performance of the algorithm is now solidly benchmarked, and shows significant improvement in peptide and protein ID over previous methods. I agree with the authors that the impact of the manuscript lies also in its significant performance gain over previous reports, in particular over shorter gradients, and the readily useful software packages for the general community. I believe the manuscript is overall suitable for publication at Nature Communications, after some minor corrections.

We thank the reviewer after appreciating the significant changes we introduced in the revision.

Minor issue:

- Given the main novelty of the work is in the introduction of a 2D peak picking algorithm, the authors should clarify this sentence in the abstract, and maybe a few other places: “We present software algorithms and the generation of optimized spectral libraries for neural network-based processing of ion mobility data acquired with dia-PASEF ...”, such that it is clear to the reader where the novelty and contribution of the current manuscript is - rather than leaving a vague impression that this work is the first one that uses neural network-based method for dia-PASEF data analysis.

We have now revised the abstract, to highlight the novel 2D-peak-picking algorithm and indicate that the workflow takes advantage of neural network-based processing, making it clear that the neural networks were developed previously.

- The 2D peak picking algorithm involves quite a few manual parameter settings, e.g. for the maximum ion mobility tolerance and m/z tolerance bounds, which are at the heart of the improvement performance claimed here. It will be helpful to the readers if the authors can add a short sentence describing how these parameters were chosen or tuned during development of the software.

We have now indicated that the parameters were chosen based on the observed peak characteristics in the data. We further indicated that the workflow might further benefit from future optimisations of these for novel acquisition schemes developed in the future.

Reviewer #2

The authors have added new analysis, benchmarks, figures and modified the main manuscript to accommodate the main aspects of the requested changes. However, some details still remain open:

The authors mentioned in their earlier rebuttal that some changes were made on the architecture of the NN-based classifier and that the peak-group scoring was refactored. None of these changes are mentioned or described in this manuscript. I welcome that DIA-NN is freely available, however, the scientific community is not able to re-implement the advances discovered and proposed by DIA-NN in their own workflows without either 1) the source code or 2) a description of the changes. At least the changes on the neural architecture need to be described.

We appreciate the interest of the Reviewer in the algorithms used, however DIA-NN is a large project consisting of many parallel developments. In this work, we described in great detail the main innovation and the core algorithm behind the ion mobility module in DIA-NN, that is the 2D-peak-picking module. Naturally, the software is being actively developed, and all other modules of the software, such as the neural network classifier, which is to significant extent responsible for the transformative improvements we show here, are being gradually improved. So while we are not able to describe all changes, we believe that the description of the new method in this manuscript should be sufficient for everyone to understand and re-implement, if desired.

I remain skeptical that the title is an appropriate description of the main contribution of the manuscript. The neural network-based classifier is one of the key components of the published DIA-NN workflow and is arguably not used for processing the (raw) IM data. It “merely” aggregates information of various scores/metrics and as such is not aware that some scores represent IM information. The aspect that an NN-classifier increases IDs was highlighted in the original publication. As the authors describe, the main (algorithmic) contribution of this manuscript is the 2D-peak-picking, which is disconnected from the NN-based classifier. While the combination may still result in better numbers, the aspect of the NN-classifier is not novel anymore. I leave the final decision to the editor, but a title like “Integration of ion mobility data into neural network-based classifier for deep proteomics of low sample amounts” seems more appropriate and would still contain the relevant buzz words.

We have now revised the title.